# Generalization Bounds for Meta-Learning: An Information-Theoretic Analysis

**Qi Chen**[*]
Université Laval

**Changjian Shui**[†]
Université Laval

**Mario Marchand**[‡]
Université Laval

## Abstract

We derive a novel information-theoretic analysis of the generalization property of meta-learning algorithms. Concretely, our analysis proposes a generic understanding of both the conventional learning-to-learn framework [1] and the modern model-agnostic meta learning (MAML) algorithms [2]. Moreover, we provide a data-dependent generalization bound for a stochastic variant of MAML, which is *non-vacuous* for deep few-shot learning. As compared to previous bounds that depend on the square norm of gradients, empirical validations on both simulated data and a well-known few-shot benchmark show that the proposed bound is orders of magnitude tighter in most situations.

## 1   Introduction

Learning a task with limited samples is crucial for real-world machine learning applications, where proper *prior knowledge* is a key component for a successful transfer. Meta-Learning [3] or learning-to-learn (LTL) aims to extract such information through previous training tasks, which has recently re-emerged as an important topic.

Modern approaches based on MAML [2] have gained tremendous success by exploiting the capabilities of deep neural networks [4–9]. However, many theoretical questions still remain elusive. For instance, in the most popular methods for *few-shot learning* [10], the task-specific parameters and meta-parameter are updated in support (also called meta-train) and query (also called meta-validation) set, respectively. However, the majority of existing theoretical results such as [1, 11–14] do not provide a formal understanding of such popular practice. Moreover, modern meta-learning approaches have incorporated over-parameterized deep neural networks, where conducting the theoretical analysis becomes even more challenging.

In this paper, we introduce a novel theoretical understanding of the generalization property of meta-learning through an information-theoretical perspective [15]. Compared with previous theoretical results, the highlights of our contributions are as follows:

**Unified Approach** We analyze two popular scenarios. 1) The conventional LTL [11], where the meta-parameters and task-specific parameters are updated within the same data set (referred as *joint training*). 2) The modern MAML-based approaches where the meta-parameters and task specific parameters are updated on distinct data sets (referred as *alternate training*), and for which the existing theoretical analysis is rare.

**Flexible Bounds** The proposed meta-generalization error bounds are highly flexible: they are algorithm-dependant, data-dependant, and are valid for non-convex loss functions. **1)** Specifically, the generalization error bound for joint-training (Theorem 5.1) is controlled by the mutual information

---

[*]Department of Computer Science and Software Engineering, `<qi.chen.1@ulaval.ca>`
[†]Department of Electrical Engineering and Computer Engineering, `<changjian.shui.1@ulaval.ca>`
[‡]Department of Computer Science and Software Engineering, `<mario.marchand@ift.ulaval.ca>`

35th Conference on Neural Information Processing Systems (NeurIPS 2021).

between the *output of the randomized algorithm* and *the whole data set*. It can cover the typical results of [12, 1], which can be interpreted with an environment-level and a task-level error. In addition, it reveals the benefit of meta learning compared to single task learning. **2)** Moreover, the generalization error bound for alternate-training (Theorem 5.2) is characterized by the conditional mutual information between the *output of the randomized algorithm* and the *meta-validation dataset*, conditioned on the *meta-train dataset*. Intuitively, when the outputs of a meta learning algorithm w.r.t. different input data-sets are similar (*i.e.* the algorithm is stable w.r.t. the data), the meta-generalization error bound will be small. This theoretical result is coherent with the recently-proposed Chaser loss in Bayes MAML [16].

**Non-vacuous bounds for gradient-based few-shot learning** Conventional gradient-based meta-learning theories heavily rely on the assumption of a Lipschitz loss. However, [17] pointed out that this Lipschitz constant for simple neural networks can be extremely large. Thus, conventional gradient-based upper bounds are often *vacuous* for deep few-shot scenarios. In contrast, we propose a tighter data-depend bound that depends on the expected *gradient-incoherence* rather than the gradient norm (the approximation of the Lipschitz constant) [18] for the Meta-SGLD algorithm, which is a stochastic variant of MAML that uses the Stochastic Gradient Langevin Dynamics (SGLD) [19]. We finally validate our theory in few-shot learning scenarios and obtain orders of magnitude tighter bounds in most situations, compared to conventional gradient-based bounds.

## 2   Related Work

**Conventional LTL** The early theoretic framework, introduced by Baxter [11], proposed the notion of *task environment* and derived uniform convergence bounds based on the capacity and covering numbers of function classes. Pentina and Lampert [12] proposed PAC-Bayes risk bounds that depend on environment-level and task-level errors. Amit and Meir [1] extended this approach and provided a tighter risk bound. However, their theory applies to stochastic neural networks and used factorized Gaussians to approximate the parameters' distributions, which is computationally expensive to use in practice. Jose and Simeone [20] first analyzed meta-learning through information-theoretic tools, while they applied the assumptions that hide some probabilistic relations and obtained theoretical results substantially different from those presented here. Limited to the space, a more detailed discussion is provided in Appendix F.

**Gradient based meta-learning** In recent years, gradient-based meta-learning such as MAML [2] have drawn increasing attention since they are model-agnostic and are easily deployed for complex tasks like reinforcement learning, computer vision, and federate learning [21–24]. Then, Reptile [25] provided a general first-order gradient calculation method. Other methods combine MAML and Bayesian methods through structured variational inference [26] and empirical Bayes [27]. In Bayes MAML[16], they propose a fast Bayesian adaption method using Stein variational gradient descent and conceived a Chaser loss which coincides with the proposed Theorem 5.2.

On the theoretical side, Denevi et al. [13] analyzed the average excess risk for Stochastic Gradient Descent (SGD) with Convex and Lipschitz loss. Balcan et al. [14] studied meta-learning through the lens of online convex optimization, and has provided a guarantee with a regret bound. Khodak et al. [28] extended to more general settings where the task-environment changes dynamically or the tasks share a certain geometric structure. Other guarantees for online meta-learning scenarios are provided by Denevi et al. [29] and Finn et al. [30]. Finally, [31, 32] also provided a convergence analysis for MAML-based methods.

**On meta train-validation split** Although the support query approaches are rather difficult to analyze, some interesting works have appeared on the simplified linear models. Denevi et al. [33] first studied train-validation split for linear centroid meta-learning. They proved a generalization bound and concluded that there exists a trade-off for train-validation split, which is consistent with Theorem 5.2 in our paper. Bai et al. [34] applied the random matrix theoretical analysis for a disentangled comparison between joint training and alternate training under the realizable assumption in linear centroid meta-learning. By calculating the closed-form concentration rates over the mean square error of parameter estimation for the two settings, they obtained a better rate constant with joint training. However, we aim to provide a generic analysis and do not make such a realizable assumption. We believe an additional excess risk analysis with more assumptions is needed for a similar comparison, which is out of the scope of this article. Moreover, Saunshi et al. [35] analyzed the train-validation

split for linear representation learning. They showed that the train-validation split encourages learning a low-rank representation. More detailed discussion and comparison can be found in Appendix F.

**Information-theoretic learning for single tasks** We use here an information-theoretic approach, introduced by Russo and Zou [36] and Xu and Raginsky [15], for characterizing single-task learning. Characterizing the generalization error of a learning algorithm in terms of the mutual information between its input and output brings the significant advantage of the ability to incorporate the dependence on the data distribution, the hypothesis space, and the learning algorithm. This is in sharp contrast with conventional VC-dimension bounds and uniform stability bounds. Tighter mutual information bounds between the parameters and a single data point are explored in [37]. Pensia et al. [38] applied the mutual-information framework to a broad class of iterative algorithms, including SGLD and stochastic gradient Hamiltonian Monte Carlo (SGHMC). Negrea et al. [39] provided data-dependent estimates of information-theoretic bounds for SGLD. For a recent comprehensive study, see Steinke and Zakynthinou [40].

# 3  Preliminaries

**Basic Notations** We use upper case letters, e.g. $X, Y$, to denote random variables and corresponding calligraphic letters $\mathcal{X}, \mathcal{Y}$ to denote the sets which they are defined on. We denote as $P_X$, the marginal probability distribution of $X$. Given the Markov chain $X \to Y$, $P_{Y|X}$ denotes the conditional distribution or the Markov transition kernel. $X \perp\!\!\!\perp Y$ means $X$ and $Y$ are independent.

And let us recall some basic definitions:

**Definition 3.1.** *Let $\psi_X(\lambda) \stackrel{def}{=} \log \mathbb{E}[e^{\lambda(X-\mathbb{E}[X])}]$ denote the cumulant generating function(CGF) of random variable $X$. Then $X$ is said to be $\sigma$-subgaussian if we have*

$$\psi_X(\lambda) \leq \frac{\lambda^2 \sigma^2}{2}, \forall \lambda \in \mathbb{R}.$$

**Definition 3.2.** *Let $X$, $Y$ and $Z$ be arbitrary random variables, and let $D_{KL}$ denote the KL divergence. The mutual information between $X$ and $Y$ is defined as:*

$$I(X;Y) \stackrel{def}{=} D_{KL}(P_{X,Y}||P_X P_Y).$$

*The disintegrated mutual information between $X$ and $Y$ given $Z$ is defined as:*

$$I^Z(X;Y) \stackrel{def}{=} D_{KL}(P_{X,Y|Z}||P_{X|Z}P_{Y|Z}).$$

*The corresponding conditional mutual information is defined as:*

$$I(X;Y|Z) \stackrel{def}{=} \mathbb{E}_Z[I^Z(X;Y)].$$

**Information theoretic bound for single task learning** We consider an unknown distribution $\mu$ on an instance space $\mathcal{Z} = \mathcal{X} \times \mathcal{Y}$, and a set of independent samples $S = \{Z_i\}_{i=1}^m$ drawn from $\mu$: $Z_i \sim \mu$ and $S \sim \mu^m$. Given a parametrized hypothesis space $\mathcal{W}$ and a loss function $\ell : \mathcal{W} \times \mathcal{Z} \to R$, the true risk and the empirical risk of $w \in \mathcal{W}$ are respectively defined as $R_\mu(w) \stackrel{def}{=} \mathbb{E}_{Z \sim \mu} \ell(w, Z)$ and $R_S(w) \stackrel{def}{=} (1/m) \sum_{i=1}^m \ell(w, Z_i)$.

Following the setting of information-theoretic learning [36, 15, 37], a learning algorithm $\mathcal{A}$ is a randomized mapping that takes a dataset $S$ as input and outputs a hypothesis $W$ according to a conditional distribution $P_{W|S}$, *i.e.*, $W = \mathcal{A}(S) \sim P_{W|S}$.[4] The (mean) generalization error $gen(\mu, \mathcal{A}) \stackrel{def}{=} \mathbb{E}_{W,S}[R_\mu(W) - R_S(W)]$ of an algorithm $\mathcal{A}$ is then bounded according to:

**Theorem 3.1.** *(Xu and Raginsky [15]) Suppose that for each $w \in \mathcal{W}$, the prediction loss $\ell(w, Z)$ is $\sigma$-subgaussian with respect to $Z \sim \mu$. Then for any randomized learner $\mathcal{A}$ characterized by $P_{W|S}$, for $S \sim \mu^m$, we have*

$$|gen(\mu, \mathcal{A})| \leq \sqrt{\frac{2\sigma^2}{m} I(W;S)}.$$

$I(W;S)$ is the mutual information between the input and output of algorithm $\mathcal{A}$ (see definition in Definition A.2). Theorem 3.1 reveals that the less the output hypothesis $W$ depends on the dataset $S$, the smaller the generalization error of the learning algorithm will be.

---

[4]Note that the conditional distribution $P_{W|S}$ is different from the posterior distribution in Bayes learning.

# 4 Problem Setup

Following [11], we assume that all tasks originate from a common *environment* $\tau$, which is a probability measure on the set of probability measures on $\mathcal{Z} = \mathcal{X} \times \mathcal{Y}$. The draw of $\mu \sim \tau$ represents encountering a learning task $\mu$ in the environment $\tau$. To run a learning algorithm for a task, we need to draw a set of data samples from $\mu$. In meta learning, there are multiple tasks, for simplicity, we assume that each task has the same sample size $m$. Based on Maurer et al. [41], the environment $\tau$ induces a mixture distribution $\mu_{m,\tau}$ on $\mathcal{Z}^m$ such that $\mu_{m,\tau}(A) = \mathbb{E}_{\mu \sim \tau}[\mu^m(A)], \forall A \subseteq \mathcal{Z}^m$. Thus the $m$ data points in $S$ that are independently sampled from a random task $\mu$ encountered in $\tau$ is denoted as $S \sim \mu_{m,\tau}$.

Consequently, for $n$ train tasks that are independently sampled from the environment $\tau$, each train data set is denoted as $S_i \sim \mu_{m,\tau}$ for $i \in [n]$. Analogously, for $k$ test tasks data sets, we denote $S_i^{\text{te}} \sim \mu_{m,\tau}$ for each $i \in [k]$. We further denote the (full) training set as $S_{1:n} = (S_1, ..., S_n)$ and the (full) testing set as $S_{1:k}^{\text{te}} = (S_1^{\text{te}}, ..., S_k^{\text{te}})$.

### Meta Learner & Base Learner

Since different tasks are assumed to be an i.i.d. sampling from $\tau$, they should share some common information. We use a *meta parameter* $U \in \mathcal{U}$ to represent this shared knowledge. We also denote by $W_{1:n} = (W_1, \ldots, W_n)$ the *task specific parameters*, where each $W_i \in \mathcal{W}, \forall i \in [n]$. By exploring the relations between $U$ and $W$, we can design different meta learning algorithms. For example, [12, 1] treated $U$ as the hyper-parameters of the base learner that produces $W$. In gradient based meta-learning such as MAML [2], $U$ was chosen to be an initialization of $W$ (hence, $\mathcal{U} = \mathcal{W}$) for a gradient-descent base learner.

We define the *meta learner* $\mathcal{A}_{\text{meta}}$ as an algorithm that takes the data sets $S_{1:n}$ as input, and then outputs a random meta-parameter $U = \mathcal{A}_{\text{meta}}(S_{1:n}) \sim P_{U|S_{1:n}}$, which is a distribution that characterizes $\mathcal{A}_{\text{meta}}$. When learning a new task, the *base learner* $\mathcal{A}_{\text{base}}$ uses a new data set $S \sim \mu_{m,\tau}$ and the estimated meta-parameter $U$ to output a stochastic predictor $W = \mathcal{A}_{\text{base}}(U, S) \sim P_{W|U,S}$.[5]

To evaluate the quality of the meta information $U$ for learning a *new task*, we define the *true meta risk*, given the base learner $\mathcal{A}_{\text{base}}$, as

$$R_\tau(U) \overset{\text{def}}{=} \mathbb{E}_{S \sim \mu_{m,\tau}} \mathbb{E}_{W \sim P_{W|S,U}}[R_\mu(W)].$$

### Joint Training & Alternate Training

Since $\tau$ and $\mu$ are unknown, we can only estimate $U$ and $W$ from the observed data. Generally, there are two different types of methods for evaluating meta and task parameters.

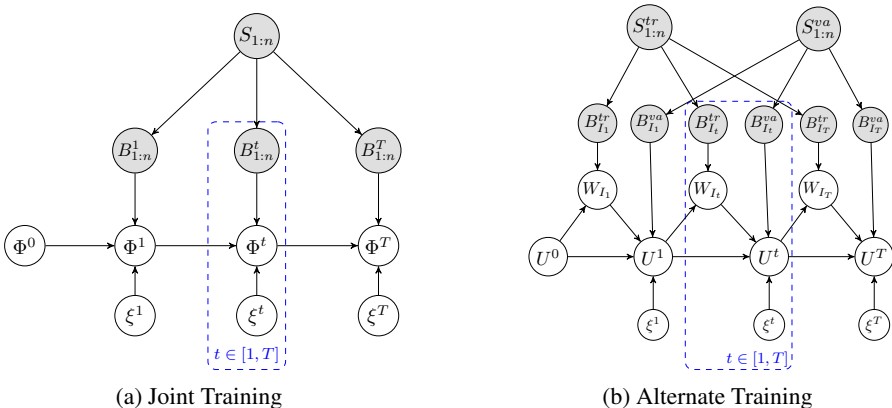

(a) Joint Training        (b) Alternate Training

Figure 1: Parameter updating strategy through noisy iterative approach.

---

[5]Although the base learner is the same, $P_{W_i|U,S_i}$ is different for each task $i$ due to the different data set $S_i$.

For *Joint Training* [1, 12], the whole dataset $S_{1:n}$ is used to jointly evaluate all the parameters $(U, W_{1:n})$ in parallel. A similar training protocol is illustrated in Fig. 1a. Then the corresponding *empirical meta risk* w.r.t. $U$ is defined as:

$$R_{S_{1:n}}(U) \stackrel{\text{def}}{=} \tfrac{1}{n} \sum_{i=1}^{n} \mathbb{E}_{W_i \sim P_{W_i | S_i, U}}[R_{S_i}(W_i)].$$

For *Alternate training*, used in modern deep meta-learning algorithms [2], $S_i$ is randomly split into two smaller datasets: a meta-train set $S_i^{\text{tr}}$ with $|S_i^{\text{tr}}| = m_{\text{tr}}$ and a meta-validation set $S_i^{\text{va}}$ with $|S_i^{\text{va}}| = m_{\text{va}}$ examples for each $i \in [n]$. In few-shot learning, $S_{1:n}^{\text{tr}}$ and $S_{1:n}^{\text{va}}$ are denoted as the support set and query set. Additionally, we have $m = m_{\text{tr}} + m_{\text{va}}$ and $S_i^{\text{tr}} \perp\!\!\!\perp S_i^{\text{va}}$. An example of the training protocol is illustrated in Fig. 1b, where $(U, W_{1:n})$ are alternately updated through $S_{1:n}^{\text{va}}$ and $S_{1:n}^{\text{tr}}$, respectively. The corresponding *empirical meta risk* w.r.t $U$ is defined as:

$$\tilde{R}_{S_{1:n}}(U) \stackrel{\text{def}}{=} \tfrac{1}{n} \sum_{i=1}^{n} \mathbb{E}_{W_i \sim P_{W_i | S_i^{\text{tr}}, U}}[R_{S_i^{\text{va}}}(W_i)]$$

Then, the *meta generalization error* within these two modes w.r.t. $\mathcal{A}_{\text{meta}}$ and $\mathcal{A}_{\text{base}}$ are respectively defined as

$$\text{gen}_{\text{meta}}^{\text{joi}}(\tau, \mathcal{A}_{\text{meta}}, \mathcal{A}_{\text{base}}) \stackrel{\text{def}}{=} \mathbb{E}_{U, S_{1:n}}[R_\tau(U) - R_{S_{1:n}}(U)],$$

$$\text{gen}_{\text{meta}}^{\text{alt}}(\tau, \mathcal{A}_{\text{meta}}, \mathcal{A}_{\text{base}}) \stackrel{\text{def}}{=} \mathbb{E}_{U, S_{1:n}}[R_\tau(U) - \tilde{R}_{S_{1:n}}(U)].$$

## 5 Information-Theoretic Generalization Bounds

We provide here novel generalization bounds for joint and alternate training, which are respectively characterized by mutual information (MI) and conditional mutual information (CMI). These theoretical results are valid for any *randomized* algorithm $\mathcal{A}_{\text{meta}}$ and $\mathcal{A}_{\text{base}}$. But for some deterministic algorithms producing deterministic predictors, the mutual information bound can be vacuous.

### 5.1 Mutual Information (MI) Bound in Joint Training

**Theorem 5.1.** *Suppose all tasks use the same loss $\ell(Z, w)$, which is $\sigma$-subgaussian for each $w \in \mathcal{W}$, where $Z \sim \mu, \mu \sim \tau$. Then, the meta generalization error for joint training is upper bounded by*

$$|\text{gen}_{\text{meta}}^{\text{joi}}(\tau, \mathcal{A}_{\text{meta}}, \mathcal{A}_{\text{base}})| \leq \sqrt{\frac{2\sigma^2}{nm} I(U, W_{1:n}; S_{1:n})}.$$

The proof of Theorem 5.1 is presented in Appendix B.1. Moreover, according to the chain rule of mutual-information, the error bound in Theorem 5.1 can be further decomposed as

$$\sqrt{\tfrac{2\sigma^2}{mn} \left( I(U; S_{1:n}) + \sum_{i=1}^{n} I(W_i; S_i | U) \right)} \leq \sqrt{\tfrac{2\sigma^2}{mn} I(U; S_{1:n})} + \sqrt{\tfrac{2\sigma^2}{mn} \sum_{i=1}^{n} I(W_i; S_i | U)}.$$

**Discussions** The first and second terms reflect, respectively, the environmental and task-level uncertainty. **1)** In the limit of a very large number of tasks ($n \to \infty$) and a finite number $m$ of samples per task, the first term converges to zero, while the second term remains non-zero. This is consistent with Theorem 1 of Bai et al. [34], where they proved that joint training has a bias in general. However, this non-zero term will be smaller than the mutual information of single-task learning. Indeed, let $I(W; S)$ denotes the mutual information of single-task learning, we have, as shown in Appendix B.2, that $I(W; S) \geq I(W; S | U) \approx \frac{1}{n} \sum_{i=1}^{n} I(W_i; S_i | U)$, which illustrates the benefits of learning the meta-parameter $U$. **2)** When we have a constant number $n$ of tasks, while the number $m$ of samples per task goes to infinity, the whole bound will converge to zero. Note that the meta generalization error bound reflects how the meta-information assists a new task to learn. If the new task has a sufficiently large number $m$ of samples, the generalization error will be small, and the meta-information $U$ does not significantly help learning the new task.

**Relation with previous work** Since mutual information implicitly depends on the unknown distribution $\tau$, it is hard to estimate and minimize [42]. By introducing an arbitrary distribution-free prior $Q$ on $\mathcal{U} \times \mathcal{W}^n$, we can upper bound $I(U, W_{1:n}; S_{1:n}) \leq I(U, W_{1:n}; S_{1:n}) + D_{\text{KL}}(P_{U, W_{1:n}} || Q) = \mathbb{E}_{S_{1:n}} D_{\text{KL}}(P_{U, W_{1:n} | S_{1:n}} || Q)$ (see Lemma A.1). If we set a joint prior $Q = \mathcal{P} \times \prod_{i=1}^{n} P$, then the bound of Theorem 5.1 becomes similar to the one proposed by [1], where $\mathcal{P}$ is the hyper-prior and $P$ is the task-prior in their settings. Finally, note that the bound of Theorem 5.1 is tighter than the one proposed by [12], where the KL divergence is outside of the square root function.

## 5.2 Conditional Mutual Information (CMI) Bound for Alternate Training

**Theorem 5.2.** *Assume that all the tasks use the same loss function $\ell(Z, w)$, which is $\sigma$-subgaussian for each $w \in \mathcal{W}$, where $Z \sim \mu, \mu \sim \tau$. Then we have*

$$|gen_{meta}^{alt}(\tau, \mathcal{A}_{meta}, \mathcal{A}_{base})| \leq \mathbb{E}_{S_{1:n}^{tr}} \sqrt{\frac{2\sigma^2 I^{S_{1:n}^{tr}}(U, W_{1:n}; S_{1:n}^{va})}{nm_{va}}} \leq \sqrt{\frac{2\sigma^2 I(U, W_{1:n}; S_{1:n}^{va}|S_{1:n}^{tr})}{nm_{va}}}.$$

See the proof in Appendix B.3. The second inequality is obtained with the Jensen's inequality for the concave square root function and Lemma A.3. Additionally, we can apply the chain rule on the conditional mutual information, to obtain the following decomposition:

$$I(U, W_{1:n}; S_{1:n}^{va}|S_{1:n}^{tr}) = I(U; S_{1:n}^{va}|S_{1:n}^{tr}) + \sum_{i=1}^{n} I(W_i; S_i^{va}|U, S_i^{tr})$$

$$= \mathbb{E}_{S_{1:n}} D_{\mathrm{KL}}(P_{U|S_{1:n}}||P_{U|S_{1:n}^{tr}}) + \mathbb{E}_{U, S_{1:n}} \sum_{i=1}^{n} D_{\mathrm{KL}}(P_{W_i|S_i, U}||P_{W_i|S_i^{tr}, U})).$$

**Discussions** The aforementioned decomposition reveals the following intuition: Suppose the outputs of the base learner and meta learner w.r.t. different input data-sets are similar (*i.e.* the learning algorithms are stable w.r.t. the data). In that case, the meta-generalization error bound will be small. Moreover, since the bound of Theorem 5.2 is data-dependent w.r.t. $S_{1:n}^{tr}$, we can obtain tighter theoretical results through these data-dependent estimates. This is to be contrasted with the mutual information bound of Theorem 5.1, which depends on the unknown distribution and can thus be inflated through the variational form. In Sec 6, we analyze noisy iterative algorithms in deep few-shot learning to obtain tighter estimates. Besides, there exists an inherent trade-off in choosing $m_{va}$. If $m_{va}$ is large, then the denominator in the bound is large. However, since $m_{va} = m - m_{tr}$, $D_{\mathrm{KL}}(P_{U|S_{1:n}}||P_{U|S_{1:n}^{tr}})$ and $D_{\mathrm{KL}}(P_{W_i|S_i, U}||P_{W_i|S_i^{tr}, U})$ will also become large since smaller $m_{tr}$ will lead to less reliable outputs.

# 6 Generalization Bounds for Noisy Iterative Algorithms

We will now exploit Theorem 5.1 and 5.2. to analyze concrete algorithms. Specifically, in noisy iterative algorithms, all the iterations are related through a Markov structure, which can naturally apply the information chain rule. Our theoretical results focus on one popular instance: SGLD [19], which is a variant of Stochastic Gradient Descent (SGD) with the addition of a scaled isotropic Gaussian noise to each gradient step. The injected noise allows SGLD to escape the local minima and asymptotically converge to global minimum for sufficiently regular non-convex objectives [43]. It is worth mentioning that other types of iterative algorithms such as SG-HMC can be also analyzed within our theoretical framework, which is left as the future work.

Since the algorithms to be analyzed require sampling mini-batches of sample at each iteration, we make the following independence assumption:

**Assumption 1** *The sampling strategy is independent of the parameters and the previous samplings.*

## 6.1 Bound for Joint Training with Bounded Gradient

In joint training, the meta and base parameters are updated simultaneously. We denote $\Phi \overset{\text{def}}{=} (U, W_{1:n}) \in \mathcal{U} \times \mathcal{W}^n, \mathcal{U} \subseteq \mathbb{R}^k, \mathcal{W} \subseteq \mathbb{R}^d$. The training strategy is illustrated in Fig 1a. Concretely, the learning algorithm executes $T$ iterations. We further denote $\Phi^t$ as the updated parameter at iteration $t \in [T]$, with $\Phi^0$ being a random initialization.

At iteration $t \in [T]$ and for task $i \in [n]$, we randomly sample a batch $B_i^t \subseteq S_i$ of size $b$ and an isotropic Gaussian noise $\xi^t \sim N(0, \sigma_t^2 \mathbb{I}_{(nd+k)})$. Let $\xi^t = (\xi_0^t, \ldots, \xi_n^t)$, where $\xi_0^t \in \mathcal{U}$, and $\xi_i^t \in \mathcal{W}, \forall i \in [n]$. Then the updating rule at iteration $t$ can be expressed as

$$\Phi^t = \Phi^{t-1} - \eta_t G(\Phi^{t-1}, B_{1:n}^t) + \xi^t,$$

where $G$ is the gradient of the empirical meta-risk on $B_{1:n}^t$ w.r.t. all the parameters $\Phi$, and where $\eta_t$ is the learning rate. In addition, we assume bounded gradients:

**Assumption 2** *The gradients are bounded,* i.e., $\sup_{\Phi \in R^{(nd+k)}, s \in \mathcal{Z}^{bn}} ||G(\Phi, s)||_2 \leq L$, *with $L > 0$.*

Then the mutual information in Theorem 5.1 can be upper-bounded as follows.

**Theorem 6.1.** *Based on Theorem 5.1, for the SGLD algorithm that satisfies Assumptions 1 & 2, the mutual information for joint training satisfies*

$$I(\Phi; S_{1:n}) \leq \sum_{t=1}^{T} \frac{nd+k}{2} \log\left(1 + \frac{\eta_t^2 L^2}{(nd+k)\sigma_t^2}\right).$$

*Specifically, if $\sigma_t = \sqrt{\eta_t}$, and $\eta_t = \frac{c}{t}$ for $c > 0$, we have:*

$$|gen_{meta}^{joi}(\tau, \mathcal{A}_{meta}, \mathcal{A}_{base})| \leq \frac{\sigma L}{\sqrt{nm}}\sqrt{c \log T + c}.$$

See the proof in Appendix B.4. It is worth mentioning that Amit and Meir [1] used $\mathcal{U} \subseteq \mathbb{R}^{2d}$, $\mathcal{W} \subseteq \mathbb{R}^d$. They adopt a similar variational form of our mutual information bound, where they use a factorized Gaussian $Q_\theta = \mathcal{N}(\theta, \mathbb{I}_{2d})$ to approximate $P_{U|S_{1:n}}$ and $Q_{\phi_i} = \mathcal{N}(\mu_i, \sigma_i^2)$ to approximate $P_{W_i|U,S_i}$. They set $P_U = \mathcal{N}(0, \mathbb{I}_{2d})$, and $P_{W_i|U} = \mathcal{N}(\mu_P, \sigma_P^2)$, where $(\mu_P, \sigma_P^2) \sim Q_\theta$. Then they optimize the meta empirical risk plus the bound w.r.t. the parameters $\theta \in \mathbb{R}^{2d}$ and $(\mu_i, \sigma_i^2) \in \mathbb{R}^{2d}$ by SGD. Our method is different since we do not use parametric approximations. Instead, we simulate the joint distribution with SGLD.

## 6.2 Bound for Alternate Training with Gradients Incoherence

The updating strategy for alternate training is illustrated in Fig. 1b. We also use SGLD for the meta learner $\mathcal{A}_{meta}$ and base learner $\mathcal{A}_{base}$, and denote this algorithm by Meta-SGLD. To build a connection with MAML, we consider the scenarios with $\mathcal{U} = \mathcal{W} \subseteq \mathbb{R}^d$, where meta-parameter $U$ is a common initialization for the task parameters $W_{1:n}$ to achieve a fast adaptation.

The Meta-SGLD algorithm has a nested loop structure: the outer loop includes $T$ iterations of SGLD for updating the meta-parameters $U$; at each outer loop iteration $t \in [T]$, there exists several parallel inner loops, where each loop is a $K$-iteration SGLD to update different task-specific parameters $W_i$.

**Outer Loop Updates**

It is computationally expensive to learn meta information from all the tasks when the number $n$ of tasks is large—a common situation in few-shot learning. Thus, for each $t \in [T]$, we sample a mini-batch of tasks that are indexed by $I_t \subseteq [n]$. Then the corresponding meta-train and meta-validation data sets are denoted as $B_{I_t}^{tr}$ and $B_{I_t}^{va}$, respectively. The task specific parameters are denoted as $W_{I_t} = \{W_i : i \in I_t\}$. In addition, an isotropic Gaussian noise $\xi^t \sim \mathcal{N}(\mathbf{0}, \sigma_t^2 \mathbb{I}_d)$ is also injected during the update. Then, the update rule w.r.t. $U$ is expressed as:

$$U^t = U^{t-1} - \eta_t \nabla \tilde{R}_{B_{I_t}^{va}}(U^{t-1}) + \xi^t,$$

where $\tilde{R}_{B_{I_t}^{va}}(U^{t-1}) = \frac{1}{|I_t|}\sum_{i \in I_t} \mathbb{E}_{W_i \sim P_{W_i|B_{i,t}^{tr}, U^{t-1}}}[R_{B_{i,t}^{va}}(W_i)]$ is the empirical meta risk evaluated on $B_{I_t}^{va}$, and $\eta_t$ is the meta learning rate at $t$. In addition, we denote the *gradient incoherence* of meta parameter $U$ at iteration $t$ as $\epsilon_t^u \stackrel{\text{def}}{=} \nabla \tilde{R}_{B_{I_t}}(U^{t-1}) - \nabla \tilde{R}_{B_{I_t}^{tr}}(U^{t-1})$.

**Inner Loop Updates**

Given the outer loop iteration $t$, for each inner iteration $k \in [K]$, we randomly sample a batch of data for task $i \in I_t$ from $B_{i,t}^{tr}$ (the $i$-th task in $B_{I_t}^{tr}$), which is denoted as $B_{i,t,k}^{tr}$. Then the update rules for the task parameters can be formulated as:

$$W_{i,t}^0 = U^{t-1}, W_{i,t}^k = W_{i,t}^{k-1} - \beta_{t,k} \nabla R_{B_{i,t,k}^{tr}}(W_{i,t}^{k-1}) + \zeta^{t,k},$$

where $\beta_{t,k}$ is the learning rate for task parameter, $\zeta^{t,k} \sim \mathcal{N}(\mathbf{0}, \sigma_{t,k}^2 \mathbb{I}_d)$ is the injected isotropic Gaussian noise (not shown in the figure) at inner iteration $k$. Analogously, we can compute the gradient incoherence w.r.t. the task parameters $W_{i,t}^k$. We first sample a batch $B_{i,t,k}$ from $B_{i,t}^{va} \bigcup B_{i,t}^{tr}$ (the union of training and validation task batches). Then the gradient incoherence of task specific parameters at the $k$-th inner update, task $i$, and outer iteration $t$ is defined as: $\epsilon_{t,i,k}^w \stackrel{\text{def}}{=} \nabla R_{B_{i,t,k}}(W_{i,t}^{k-1}) - \nabla R_{B_{i,t,k}^{tr}}(W_{i,t}^{k-1})$.

*Relation to MAML* Without the noise injection, the whole updating protocol described above is exactly MAML. Specifically, if we set $K = 1$, the empirical meta loss can be expressed as:

$\tilde{R}_{B_{I_t}^{va}}(U^{t-1}) = \frac{1}{|I_t|}\sum_{i\in I_t} R_{B_{i,t}^{va}}(W_{i,t}^K) = \frac{1}{|I_t|}\sum_{i\in I_t} R_{B_{i,t}^{va}}(U^{t-1} - \beta_{t,1}\nabla R_{B_{i,t,1}^{tr}}(U^{t-1}))$, and $\frac{1}{|I_t|}\sum_{i\in I_t}\nabla R_{B_{i,t}^{va}}(W_{i,t}^K)$ is the first-order MAML gradient.

Based on the nested loop structure and the independent sampling strategy, we have the following data-dependent generalization-error bound for Meta-SGLD.

**Theorem 6.2.** *Based on Theorem 5.2, for the Meta-SGLD that satisfies Assumption 1, if we set $\sigma_t = \sqrt{2\eta_t/\gamma_t}$, $\sigma_{t,k} = \sqrt{2\beta_{t,k}/\gamma_{t,k}}$, where $\gamma_t$ and $\gamma_{t,k}$ are the inverse temperatures. The meta generalization error for alternate training satisfies*

$$|gen_{meta}^{alt}(\tau, SGLD, SGLD)| \leq \sqrt{\frac{2\sigma^2 I(U, W_{1:n}; S_{1:n}^{va}|S_{1:n}^{tr})}{nm_{va}}} \leq \frac{\sigma}{\sqrt{nm_{va}}}\sqrt{\epsilon_U + \epsilon_W},$$

*where*

$$\epsilon_U = \sum_{t=1}^{T}\mathbb{E}_{B_{I_t}^{va}, B_{I_t}^{tr}, W_{I_t}, U^{t-1}}\frac{\eta_t\gamma_t\|\epsilon_t^u\|_2^2}{2}, \quad \epsilon_W = \sum_{t=1}^{T}\sum_{i=1}^{|I_t|}\sum_{k=1}^{K}\mathbb{E}_{B_{i,t,k}^{va}, B_{i,t,k}^{tr}, W_{i,t}^{k-1}}\frac{\beta_{t,k}\gamma_{t,k}\|\epsilon_{t,i,k}^w\|_2^2}{2}.$$

The proof is provided in Appendix B.5. The bound of Theorem 6.2 consists of two parts: $\epsilon_U$, which reflects the generalization error bound of the meta learner, and $\epsilon_W$, which reflects the generalization bound of the base learner. Moreover, $\epsilon_U$ and $\epsilon_W$ are characterized by the accumulated *gradient incoherence* and predefined constants such as learning rates, inverse temperatures, and number of iterations. Compared with previous works such as [13, 30], Theorem 6.2 exploits the gradient difference between two batches rather than the Lipschitz constant of the loss function (and, consequently, its tighter estimation, the gradient norm of the empirical meta-risk and the individual task risks). This can give a more realistic generalization error bound since the Lipschitz constant for neural networks is often very large [17]. In contrast, our empirical results reveal (see Sec 7) that the gradient incoherence can be much smaller than the gradient norm on average. Note that we have chosen fixed and large inverse temperatures to ensure small injected noise variance from the beginning of training. In addition, the step sizes also affect the bound w.r.t. training iteration numbers $T, K$. For example, assuming that the gradient incoherence is bounded, if we choose $\eta_t = \frac{1}{t}, \beta_{t,k} = \frac{1}{tk}$, the meta generalization error bound is in $\mathcal{O}(\sqrt{c_1\log T + c_2\log K})$, where $c_1, c_2$ are some constants. In contrast, when learning rates are fixed, the bound is in $\mathcal{O}(\sqrt{c_1 T + c_2 TK})$.

## 7 Empirical Validations

We validate Theorem 6.2 on both synthetic and real data. The numerical results demonstrate that, in most situations, the *gradient incoherence* based bound is orders of magnitude tighter than the conventional meta learning bounds with the Lipschitz assumption, which is estimated with gradient norms.[6]

### 7.1 Synthetic Data

We consider a simple example of 2D mean estimation to illustrate the meta-learning setup. We assume that the environment $\tau$ is a truncated 2D Gaussian distribution $\mathcal{N}((-4, -4)^T, 5\mathbb{I}_2)$. A new task is also defined as a 2D Gaussian $\mathcal{N}(\mu, 0.1\mathbb{I}_2)$ with $\mu \sim \tau$. To generate few-shot tasks, we sample $n = 20000$ tasks from the environment with $\mu_i \sim \tau, \forall i \in [n]$. After sampling $\mu_i$ for each task, we further sample $m = 16$ data points from $\mathcal{N}(\mu_i, 0.1\mathbb{I}_2)$. At each iteration $t$, we randomly choose a subset of 5 tasks ($|I_t| = 5$) from the whole data set. We evaluate on three different few-shot settings with $m_{va} = \{1, 8, 15\}$ and the corresponding train size $m_{tr} = \{15, 8, 1\}$. The detailed experiment setting is in Appendix E.1.

The estimated meta-generalization upper bounds are shown in Fig. 2. For a better understanding of the generalization behaviour w.r.t. the meta learner and base learner, we separately show the estimated bounds of $\sigma\sqrt{\frac{\epsilon_U}{nm_{va}}}$ (Fig. 2(a)) and $\sigma\sqrt{\frac{\epsilon_W}{nm_{va}}}$ (Fig. 2(b)). The expectation terms within the bound are estimated via Monte-Carlo sampling. To compare the conventional Lipschiz bound with

---

[6]Code is available at: `https://github.com/livreQ/meta-sgld`.

ours, we approximately calculated a tighter estimation of the bound with the expected gradient norm at each iteration [18] instead of using a fixed Lipschitz constant, which is extremely vacuous in deep learning. The other components remain the same as the gradient incoherence bound.

The results on the synthetic data set reveal a substantial theoretical benefit compared with the conventional Lipschitz bound. Specifically, the magnitude of the bound is improved by a factor of 10 to 100. Interestingly, the gap between the gradient-norm and the gradient-incoherence bound is smallest when $m_{\text{va}} = 15$. These theoretical results reveal that the generalization bound is unavoidably large if the base learner is trained on extremely few data (*e.g.*, a 1-shot scenario). Since too few train data (small $m_{\text{tr}}$) induces high randomness and large instability in each training task.

To further validate Theorem 6.2, We calculated the actual generalization gap by evaluating the expected difference between the train loss and test loss for the above mentioned tree settings. The actual generalization gap of $m_{\text{tr}} = 1$ is also much larger compared to the other two setting, which also demonstrated the instability for extreme few shot learning (See Table 1, 2 and 3 in Appendix D.1).

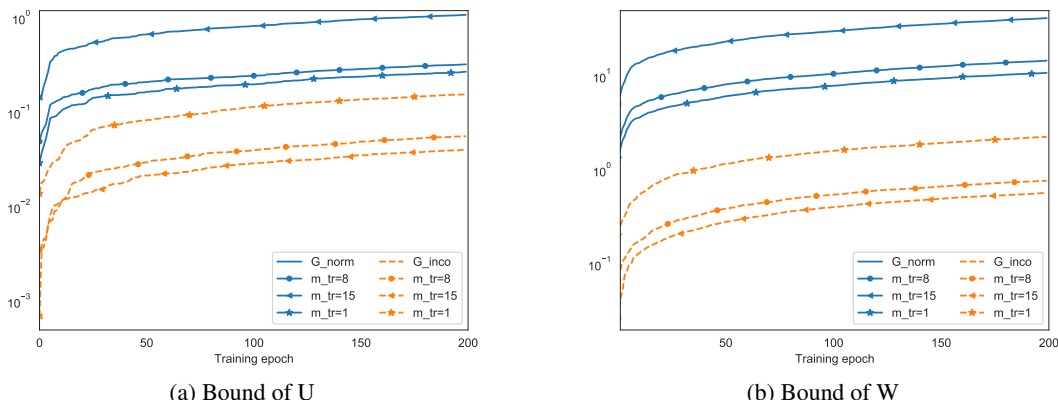

(a) Bound of U          (b) Bound of W

Figure 2: Synthetic data: Estimation of the generalization-error bound during the training ($T = 200$). *Left* Estimated error bound w.r.t. meta-learner. *Right* The estimated error bound w.r.t. the base-learner. The curves in Blue solid line and Orange dashed line represent the estimated bound through gradient-norm (G_Norm) and gradient-incoherence (G_Inco) in different few-shot settings.

## 7.2 Few-Shot Benchmark

To evaluate the proposed bound in modern deep few-shot learning scenarios, we have tested the Meta-SGLD algorithm on the Omniglot dataset [44]. The Omniglot dataset contains 1623 characters for 50 different alphabets, and each character is present in 20 instances. We followed the experimental protocol of [45, 2], which aims to learn a N-way classification task for 1-shot or 5-shot learning. In our experiment, we conducted a 5-way classification learning. A train task consists of five classes (characters) randomly chosen from the first 1200 characters, each class has $m = 16$ samples selected from the 20 instances. Similarly, a test task contains five classes randomly sampled from the rest 423 characters. Therefore, the meta train set has $n = \binom{1200}{5}$ tasks. At each epoch, we have trained the model with $|I_t| = 32$ tasks. Analogous to the simulated data, we have conducted our experiment with $m_{\text{tr}} = \{15, 8, 1\}$ and $m_{\text{va}} = \{1, 8, 15\}$ and separately visualized the two components of the bound. The detailed experimental setting is provided in Appendix E.2.

The estimated bounds are shown in Fig. 3. Analogous to the results on synthetic data, the estimated error bound trough gradient-incoherence is *tighter* than the gradient-norm based bound when $m_{\text{tr}} = 8, 15$. In particular, the gradient-incoherence bound w.r.t. $U$ is much tighter than the gradient-norm bound when $m_{\text{tr}} = 15$, which illustrates the benefits of the proposed theory. Simultaneously, the gradient-incoherence bound is similar to the gradient-norm bound when $m_{\text{tr}} = 1$, illustrating a theoretical limitation of learning with very few meta-train samples. Moreover, we observe that the optimal values for $m_{\text{va}}$ depends on the environment since the tightest bound for Omniglot is achieved with $m_{\text{va}} = 8$, which is different from what we have found for the synthetic data.

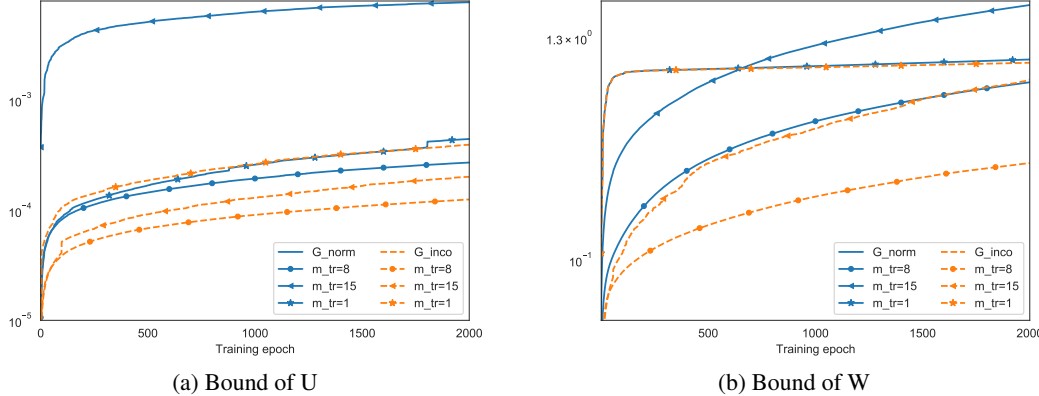

|  | (a) Bound of U | (b) Bound of W |

Figure 3: Omniglot: Estimation of the generalization-error bound during the training ($T = 2000$). The curves in Blue solid line and Orange dashed line represent the estimated bound through gradient-norm (G_Norm) and gradient-incoherence (G_Inco) in different few-shot settings

Finally, we observed that the component of the generalization error bound that originates from task-specific parameters is numerically larger than the one the originates from the meta parameter, has compared to the results for simulated data. This perhaps illustrates an inherent difficulty in learning few-shot tasks with high-dimensional and complex data sets, where estimating the generalization error bound is apparently more challenging. Additional experimental results for test accuracy comparison with MAML on the aforementioned tree settings are presented in Appendix D.2 Table 4. Comparison of bound values with the observed generalization error is also included (See Table 5,6 and 7). We believe the less evident improvement with gradient incoherence bound compared to Synthetic data can be ascribed to the utilization of Batch Normalization.

# 8 Conclusion

We derived a novel information-theoretic analysis of the generalization property of meta-learning and provided algorithm-dependent generalization error bounds for both joint training and alternate training. Compared to previous gradient-based bounds that depend on the square norm of gradients, empirical validations on both simulated data and a few-shot benchmark show that the proposed bound is orders of magnitude tighter in most situations. Finally, we think that these theoretical results can inspire new algorithms through a deeper exploration of the relation between meta-parameters and task-parameters.

## Acknowledgments and Disclosure of Funding

Work partly supported by NSERC Discovery Grant RGPIN-2016-05942 and the China Scholarship Council. We also thank SSQ Assurances and NSERC for their financial support through the Collaborative Research and Development Grant CRDPJ 529584 - 18.

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
