# A  Technical Lemmas

**Lemma A.1** (Variational Form of Mutual Information). *Let $X$ and $Y$ be two random variables. For all probability measures $Q$ defined on the space of $X$, we have*

$$I(X;Y) \leq E_Y[D_{KL}(P_{X|Y}||Q)],$$

*with equality for $Q = P_X$.*

*Proof.*

$$I(X;Y) + D_{\text{KL}}(P_X||Q)$$
$$= \iint p(x,y) \log \frac{p(x,y)}{p(x)p(y)} dxdy + \int p(x) \log \frac{p(x)}{q(x)} dx$$
$$= \iint p(x,y) \log \frac{p(x,y)}{p(x)p(y)} dxdy + \iint p(x,y) \log \frac{p(x)}{q(x)} dxdy$$
$$= \iint p(x,y) \log \frac{p(x|y)}{q(x)} dxdy$$
$$= \mathbb{E}_Y[D_{\text{KL}}(P(X|Y)||Q)].$$

Since $D_{\text{KL}}(P_X||Q) \geq 0$, the equality exists only when $Q = P_X$, which concludes the proof. $\square$

**Lemma A.2.** *Let $X, Y, Z$ be random variables. For all $\mathcal{Z}$-measurable probability measures $Q$ on the space of $X$, $I^Z(X;Y) \leq E_{Y|Z}[D_{KL}(P_{X|Y,Z}||Q)]$, with equality for $Q = P_{X|Z}$.*

*Proof.*

$$I^Z(X;Y) + D_{\text{KL}}(P_{X|Z}||Q)$$
$$= \iint p(x,y|z) \log \frac{p(x,y|z)}{p(x|z)p(y|z)} dxdy + \iint p(x|z) \log \frac{p(x|z)}{q(x)} dx$$
$$= \iint p(x,y|z) \log \frac{p(x,y|z)}{p(x|z)p(y|z)} dxdy$$
$$\quad + \iint p(x,y|z) \log \frac{p(x|z)}{q(x)} dxdy$$
$$= \iint p(x,y|z) \log \frac{p(x|y,z)}{q(x)} dxdy$$
$$= \mathbb{E}_{Y|Z}[D_{\text{KL}}(P(X|Y,Z)||Q)]$$

Since $D_{\text{KL}}(P_{X|Z}||Q) \geq 0$, the equality exists only when $Q = P_{X|Z}$, which concludes the proof. $\square$

**Lemma A.3.** *Let $X, Y, Z$ be random variables. For all $\mathcal{Z}$-measurable probability measures $Q$ defined on the space of $X$, $I(X;Y|Z) = \mathbb{E}_Z[I^Z(X;Y)] \leq E_{Y,Z}[D_{KL}(P_{X|Y,Z}||Q)]$, with equality for $Q = P_{X|Z}$.*

*Proof.* Take the expectation on the inequality of Lemma B.2 to obtain the result. $\square$

**Lemma A.4.** *(Donsker-Varadhan representation[Corollary 4.15[46]]) Let $P$ and $Q$ be two probability measures defined on a set $\mathcal{X}$. Let $g : \mathcal{X} \rightarrow R$ be a measurable function, and let $\mathbb{E}_{x \sim Q}[\exp g(x)] \leq \infty$. Then*

$$D_{KL}(P||Q) = \sup_g \{\mathbb{E}_{x \sim P}[g(x)] - \log \mathbb{E}_{x \sim Q}[\exp g(x)]\}.$$

**Lemma A.5.** *(Decoupling Estimate[Xu and Raginsky [15]]) Consider a pair of random variables $X$ and $Y$ with joint distribution $P_{X,Y}$, let $\tilde{X}$ be an independent copy of $X$, and $\tilde{Y}$ an independent copy of $Y$, such that $P_{\tilde{X},\tilde{Y}} = P_X P_Y$. For arbitrary real-valued function $f : \mathcal{X} \times \mathcal{Y} \rightarrow \mathbb{R}$, if $f(\tilde{X}, \tilde{Y})$ is $\sigma$-subgaussian under $P_{\tilde{X},\tilde{Y}}$, then:*

$$|\mathbb{E}[f(X,Y)] - \mathbb{E}[f(\tilde{X}, \tilde{Y})]| \leq \sqrt{2\sigma^2 I(X;Y)}$$

**Lemma A.6.** *Let $Q$ be an arbitrary distribution on $\mathcal{W}$, and let $S$ be an arbitrary sample of examples. The solution to the optimization problem*

$$P^* = \arg\inf_P \left\{ \mathbb{E}_{W \sim P}[R_S(W)] + \frac{1}{\beta} D_{KL}(P||Q) \right\}.$$

*is given by the Gibbs distribution*

$$dP^*(w) = \frac{e^{-\beta R_S(w)} dQ(w)}{\mathbb{E}_{W \sim Q} e^{-\beta R_S(W)}}.$$

**Lemma A.7.** *(Data Processing Inequality) Given random variables $X, Y, Z, V$, and the Markov Chain:*

$$X \to Y \to Z,$$

*then we have*

$$I(X; Z) \leq I(X; Y), I(X; Z) \leq I(Y; Z).$$

*For Markov chain*

$$V \to X \to Y \to Z,$$

*we have*

$$I(X; Z|V) \leq I(X; Y|V), I(X; Z|V) \leq I(Y; Z|V)$$

*Proof.* Since

$$I(X; Y, Z) = I(X; Z) + I(X; Y|Z) = I(X; Y) + I(X; Z|Y),$$

and with the Markov Chain, we have $X \perp\!\!\!\perp Z|Y$, therefore

$$I(X; Z|Y) = H(X|Y) - H(X|Y, Z) = 0.$$

In addition, $I(X; Y|Z) \geq 0$, so $I(X; Z) \leq I(X; Y)$.

$$I(Z; X, Y) = I(Z; X) + I(Z; Y|X) = I(Z; Y) + I(Z; X|Y) = I(Y : Z),$$

with $I(Y; Z|X) \geq 0$, we have $I(X; Z) \leq I(Y; Z)$.

Similarly, for the second Markov chain, we have $X \perp\!\!\!\perp Z|Y, V$, therefore

$$I(X; Z|Y, V) = H(X|Y, V) - H(X|Y, Z, V) = 0.$$

$$I(X; Y, Z|V) = I(X; Z|V) + I(X; Y|V, Z) = I(X; Y|V) + I(X; Z|Y, V) = I(X; Y|V)$$

So we have $I(X; Z|V) \leq I(X; Y|V)$, the rest proof is similar and omitted. $\square$

**Lemma A.8.** *Given random variables $X, Y, Z_1, Z_2$, and the graph model:*

$$Z_1 \to Z_2 \to X \leftarrow Y,$$

*then we have*

$$I(X; Y|Z_1) \leq I(X; Y|Z_2)$$

*Proof.* Apply chain rule, we get:

$$I(X; Y, Z_2|Z_1) = I(X; Y|Z_1) + I(X; Z_2|Y, Z_1) = I(X; Z_2|Z_1) + I(X; Y|Z_2, Z_1)$$

From the graph model, we have $Y \perp\!\!\!\perp Z_1$, $Y \perp\!\!\!\perp Z_2$ and $(X, Y) \perp\!\!\!\perp Z_1|Z_2$. Hence

$$I(X; Y|Z_2, Z_1) = H(X|Z_2, Z_1) - H(X|Y, Z_2, Z_1) = H(X|Z_2) - H(X|Y, Z_2) = I(X; Y|Z_2)$$

Moreover,

$$\begin{aligned} I(X, Y; Z_2|Z_1) &= I(X; Z_2|Z_1) + I(Y; Z_2|X, Z_1) \\ &= I(Y; Z_2|Z_1) + I(Z_2; X|Y, Z_1) \\ &= I(Z_2; X|Y, Z_1) \end{aligned}$$

the last equality is obtained with $Y \perp\!\!\!\perp Z_2$ and $Y \perp\!\!\!\perp Z_1$, since $I(Y; Z_2|X, Z_1) \geq 0$, we get $I(X; Z_2|Z_1) \leq I(X; Z_2|Y, Z_1)$. Consequently, we have $I(X; Y|Z_1) \leq I(X; Y|Z_2)$, conclude the proof. $\square$

# B Proof

## B.1 Proof of Theorem 5.1

**Theorem** (Meta-generalization error bound for joint training). *Suppose all tasks use the same loss* $\ell(Z, w)$*, which is* $\sigma$*-subgaussian for any* $w \in \mathcal{W}$*, where* $Z \sim \mu, \mu \sim \tau$*.Then, the meta generalization error for joint training is upper bounded by*

$$|gen_{meta}^{joi}(\tau, \mathcal{A}_{meta}, \mathcal{A}_{base})| \leq \sqrt{\frac{2\sigma^2}{nm} I(U, W_{1:n}; S_{1:n})} \,.$$

*Proof.* In contrast to previous works [12, 1, 20], which separately bound the environment-level and task-level error and then combine the two terms, we consider $U, W_{1:n}$ as a collection and directly bound the whole term. By using the chain rule for mutual information, the final result can then be split into an environment-level and a task-level contribution.

Similar to Lemma 2.5, let $\Phi = (U, W_{1:n}) \in \mathcal{U} \times \mathcal{W}^n$ be a collection of random variables such that $\Phi \not\perp S_{1:n}$, and let $\tilde{\Phi} = (\tilde{U}, \tilde{W}_{1:n}) \in \mathcal{U} \times \mathcal{W}^n$ be an in dependant copy of $\Phi$ such that $\tilde{\Phi} \perp\!\!\!\perp S_{1:n}$, *i.e.*, $\tilde{\Phi}$ is distributed according to $P_{U,W_{1:n}} = \mathbb{E}_{S_{1:n}} P_{U,W_{1:n}|S_{1:n}}$. Let

$$f(\Phi, S_{1:n}) \stackrel{\text{def}}{=} \frac{1}{n} \sum_{i=1}^{n} [R_{S_i}(W_i)] = \frac{1}{n} \sum_{i=1}^{n} \frac{1}{m} \sum_{j=1}^{m} \ell(W_i, Z_{i,j}).$$

For any $\lambda \in \mathbb{R}$, let

$$\psi_{\tilde{\Phi}, S_{1:n}}(\lambda) \stackrel{\text{def}}{=} \log \mathbb{E}_{\tilde{\Phi}, S_{1:n}} \left[ e^{\lambda(f(\tilde{\Phi}, S_{1:n}) - \mathbb{E}[f(\tilde{\Phi}, S_{1:n})])} \right]$$

$$= \log \mathbb{E}_{\tilde{\Phi}, S_{1:n}} [e^{\lambda f(\tilde{\Phi}, S_{1:n})}] - \lambda \mathbb{E}_{\tilde{\Phi}, S_{1:n}} [f(\tilde{\Phi}, S_{1:n})] \,.$$

Moreover,

$$I(\Phi; S_{1:n}) = D_{\text{KL}}(P_{\Phi, S_{1:n}} || P_\Phi P_{S_{1:n}})$$

$$= \sup_g \left\{ \mathbb{E}_{\Phi, S_{1:n}} [g(\Phi, S_{1:n})] - \log \mathbb{E}_{\tilde{\Phi}, S_{1:n}} [e^{g(\tilde{\Phi}, S_{1:n})}] \right\}$$

$$\geq \lambda \mathbb{E}_{\Phi, S_{1:n}} [f(\Phi, S_{1:n})] - \log \mathbb{E}_{\tilde{\Phi}, S_{1:n}} [e^{\lambda f(\tilde{\Phi}, S_{1:n})}], \quad \forall \lambda \in \mathbb{R}$$

$$= \lambda \mathbb{E}_{U, W_{1:n}, S_{1:n}} [f(\Phi, S_{1:n})] - \lambda \mathbb{E}_{\tilde{U}, \tilde{W}_{1:n}, S_{1:n}} [f(\tilde{\Phi}, S_{1:n})] - \psi_{\tilde{\Phi}, S_{1:n}}(\lambda)$$

$$= \lambda \mathbb{E}_{U, W_{1:n}, S_{1:n}} \frac{1}{n} \sum_{i=1}^{n} [R_{S_i}(W_i)] - \lambda \mathbb{E}_{\tilde{U}, \tilde{W}_{1:n}, S_{1:n}} \frac{1}{n} \sum_{i=1}^{n} [R_{S_i}(\tilde{W}_i)] - \psi_{\tilde{\Phi}, S_{1:n}}(\lambda) \quad (1)$$

Since $(W_i, S_i), i = 1, ..., n$ are mutually independent given $U$, and $S_1, ...S_n$ are independent, we have $p(w_{1:n}|s_{1:n}, u) = \prod_{i=1}^{n} p(w_i|s_i, u)$. Hence

$$\lambda \mathbb{E}_{U, W_{1:n}, S_{1:n}} \frac{1}{n} \sum_{i=1}^{n} [R_{S_i}(W_i)] = \lambda \mathbb{E}_{U, S_{1:n}} \frac{1}{n} \sum_{i=1}^{n} \mathbb{E}_{W_i|S_i, U} [R_{S_i}(W_i)]$$

$$= \lambda \mathbb{E}_{U, S_{1:n}} [R_{S_{1:n}}(U)] \quad (2)$$

Since $\tilde{\Phi} \perp\!\!\!\perp S_{1:n}$, we have that $P_{\tilde{W}_{1:n}|S_{1:n}, \tilde{U}} = P_{\tilde{W}_{1:n}|\tilde{U}}$ and $P_{\tilde{W}_{1:n}, S_{1:n}, \tilde{U}} = P_{\tilde{W}_{1:n}, \tilde{U}} P_{S_{1:n}}$. Hence,

$$R_\tau(\tilde{U}) = \mathbb{E}_{S \sim \mu_{m,\tau}} \mathbb{E}_{\tilde{W} \sim P_{\tilde{W}|S, \tilde{U}}} [R_\mu(\tilde{W})] = \mathbb{E}_{\mu \sim \tau} \mathbb{E}_{S|\mu \sim \mu^m} \mathbb{E}_{\tilde{W} \sim P_{\tilde{W}|\tilde{U}}} [R_\mu(\tilde{W})]$$

$$= \mathbb{E}_{\mu \sim \tau} \mathbb{E}_{\tilde{W} \sim P_{\tilde{W}|\tilde{U}}} [R_\mu(\tilde{W})] \,.$$

Therefore

$$\lambda \mathbb{E}_{\tilde{U}, \tilde{W}_{1:n}, S_{1:n}} \frac{1}{n} \sum_{i=1}^{n} R_{S_i}(\tilde{W}_i) = \lambda \mathbb{E}_{\tilde{U}, \tilde{W}_{1:n}} \mathbb{E}_{S_{1:n} \sim \mu_{m,\tau}^n} \left[ \frac{1}{n} \sum_{i=1}^{n} R_{S_i}(\tilde{W}_i) \right]$$

$$= \lambda \mathbb{E}_{\tilde{U}, \tilde{W}_{1:n}} \left[ \frac{1}{n} \sum_{i=1}^{n} \mathbb{E}_{S_i \sim \mu_{m,\tau}} R_{S_i}(\tilde{W}_i) \right]$$

$$= \lambda \mathbb{E}_{\tilde{U}, \tilde{W}_{1:n}} \left[ \frac{1}{n} \sum_{i=1}^{n} \mathbb{E}_{\mu_i \sim \tau} \mathbb{E}_{S_i | \mu_i \sim \mu_i^m} R_{S_i}(\tilde{W}_i) \right]$$

$$= \lambda \mathbb{E}_{\tilde{U}, \tilde{W}_{1:n}} \left[ \frac{1}{n} \sum_{i=1}^{n} \mathbb{E}_{\mu_i \sim \tau} \left[ \frac{1}{m} \sum_{j=1}^{m} \mathbb{E}_{Z_{i,j} \sim \mu_i} \ell(\tilde{W}_i, Z_{i,j}) \right] \right]$$

$$= \lambda \mathbb{E}_{\tilde{U}} \mathbb{E}_{\tilde{W}_{1:n} | \tilde{U}} \left[ \frac{1}{n} \sum_{i=1}^{n} \mathbb{E}_{\mu_i \sim \tau} \left[ \frac{1}{m} \sum_{j=1}^{m} \mathbb{E}_{Z_{i,j} \sim \mu_i} \ell(\tilde{W}_i, Z_{i,j}) \right] \right]$$

$$= \lambda \mathbb{E}_{\tilde{U}} \left[ \frac{1}{n} \sum_{i=1}^{n} \mathbb{E}_{\tilde{W}_i | \tilde{U}} \mathbb{E}_{\mu_i \sim \tau} R_{\mu_i}(\tilde{W}_i) \right]$$

$$= \lambda \mathbb{E}_{\tilde{U}} \mathbb{E}_{\mu \sim \tau} \mathbb{E}_{\tilde{W} | \tilde{U}} R_\mu(\tilde{W}) = \lambda \mathbb{E}_{\tilde{U}} R_\tau(\tilde{U})$$

$$= \lambda \mathbb{E}_{U, S_{1:n}} R_\tau(U) . \tag{3}$$

If we use Equations (2) and (3), then Equation (1) becomes

$$-\lambda \mathbb{E}_{U, S_{1:n}} \left[ R_\tau(U) - R_{S_{1:n}}(U) \right] \le I(\Phi; S_{1:n}) + \psi_{\tilde{\Phi}, S_{1:n}}(\lambda) , \quad \forall \lambda \in \mathbb{R} . \tag{4}$$

Since this inequality is also valid when $\lambda$ is negative, this implies that we also have

$$\mathbb{E}_{U, S_{1:n}} \left[ R_\tau(U) - R_{S_{1:n}}(U) \right] \le \frac{1}{\lambda} \left[ I(\Phi; S_{1:n}) + \psi_{\tilde{\Phi}, S_{1:n}}(-\lambda) \right] , \quad \forall \lambda > 0 .$$

Consequently,

$$\mathrm{gen}_{\mathrm{meta}}^{\mathrm{joi}}(\tau, \mathcal{A}_{\mathrm{meta}}, \mathcal{A}_{\mathrm{base}}) \le \frac{1}{\lambda} \left[ I(\Phi; S_{1:n}) + \psi_{\tilde{\Phi}, S_{1:n}}(-\lambda) \right] , \quad \forall \lambda > 0 .$$

Since $\ell(\tilde{W}, Z)$ is $\sigma$-subgaussian, we have that $f(\tilde{\Phi}, S_{1:n}) = \frac{1}{n} \sum_{i=1}^{n} \frac{1}{m} \sum_{j=1}^{m} \ell(\tilde{W}_i, Z_{i,j})$ is $\frac{\sigma}{\sqrt{nm}}$-subgaussian.[5] Hence,

$$\psi_{\tilde{\Phi}, S_{1:n}}(\lambda) \le \frac{\lambda^2 \sigma^2}{2nm} \quad \forall \lambda \in \mathbb{R} .$$

Thus, we have

$$\mathrm{gen}_{\mathrm{meta}}^{\mathrm{joi}}(\tau, \mathcal{A}_{\mathrm{meta}}, \mathcal{A}_{\mathrm{base}}) \le \frac{I(\Phi; S_{1:n})}{\lambda} + \frac{\lambda \sigma^2}{2nm} , \quad \forall \lambda > 0 .$$

By using the value of $\lambda$ that minimizes the r.h.s. of the above equation, we have

$$\mathrm{gen}_{\mathrm{meta}}^{\mathrm{joi}}(\tau, \mathcal{A}_{\mathrm{meta}}, \mathcal{A}_{\mathrm{base}}) \le \sqrt{\frac{2\sigma^2 I(\Phi; S_{1:n})}{nm}} . \tag{5}$$

Returning to Equation (4), we have for $\lambda > 0$:

$$\mathbb{E}_{U, S_{1:n}} \left[ R_\tau(U) - R_{S_{1:n}}(U) \right] \ge -\frac{1}{\lambda} \left[ I(\Phi; S_{1:n}) + \psi_{\tilde{\Phi}, S_{1:n}}(\lambda) \right] \ge -\sqrt{\frac{2\sigma^2 I(\Phi; S_{1:n})}{nm}} .$$

Hence, we also have

$$\mathrm{gen}_{\mathrm{meta}}^{\mathrm{joi}}(\tau, \mathcal{A}_{\mathrm{meta}}, \mathcal{A}_{\mathrm{base}}) \ge -\sqrt{\frac{2\sigma^2 I(\Phi; S_{1:n})}{nm}} . \tag{6}$$

Then, Equations (5) and (6) together imply that

$$\left| \mathrm{gen}_{\mathrm{meta}}^{\mathrm{joi}}(\tau, \mathcal{A}_{\mathrm{meta}}, \mathcal{A}_{\mathrm{base}}) \right| \le \sqrt{\frac{2\sigma^2 I(\Phi; S_{1:n})}{nm}} ,$$

which gives the theorem. □

---

[5]More discussion on subgaussianity can be found in Section C.

## B.2 Benefits of Meta Learning

The task specific empirical risk $R_S(W)$ is independent of the meta parameter $U$, given the task specific parameter $W$, which gives the implicit independence assumption $S \perp\!\!\!\perp U|W$. We thus have $I(U; S|W) = 0$, and the following two possible decompositions:

$$I(S; U, W) = I(W; S|U) + I(U; S) = I(W; S) + I(U; S|W) = I(W; S).$$

Since $I(U; S) \geq 0$, we obtain $I(W; S|U) \leq I(W; S)$.

As mentioned in the main paper, Theorem 5.1 can cover the PAC Bayes bound of Amit and Meir [1] with the variational form of mutual information. Their work has built a connection between PAC Bayes meta-learning and Hierarchical Variational Bayes. In Appendix A.3 of [1], they give the generative graph model for meta learning where $U \to W \to S$ (their notation used $\psi$ instead of $U$). They assumed that $S$ is independent of $U$ given $W$, in Bayes learning, this implies that $p(S|W, U) = p(S|W)$. Based on the graph model, they obtained a similar optimization objective as their PAC-Bayes meta learning algorithm, which minimizes the expected empirical risk plus the PAC Bayes bound. Germain et al. [47] has given a more obvious connection between PAC Bayes learning and Bayes learning, where optimizing the PAC Bayes bound together with the expected empirical risk gives the so called Gibbs algorithm (see Lemma A.6). When using the negative log loss, i.e., $R_S(W) = -\frac{1}{m} \log p(S|W) = -\frac{1}{m} \sum_{i=1}^{m} \log p(Z_i|W)$, the output of Gibbs algorithm coincides with the Bayes Posterior. Therefore, without the independence assumption, $R_S(W)$ should be defined as $R_S(W, U)$, which corresponds to $-\frac{1}{m} \log p(S|W, U)$ in Bayes learning.

## B.3 Proof of Theorem 5.2

**Theorem** (Meta-generalization error bound for alternate training). *Suppose all tasks use the same loss $\ell(Z, w)$, which is $\sigma$-subgaussian for any $w \in \mathcal{W}$, where $Z \sim \mu, \mu \sim \tau$. Then we have*

$$|gen_{meta}^{alt}(\tau, \mathcal{A}_{meta}, \mathcal{A}_{base})| \leq \mathbb{E}_{S_{1:n}^{tr}} \sqrt{\frac{2\sigma^2 I^{S_{1:n}^{tr}}(U, W_{1:n}; S_{1:n}^{va})}{nm_{va}}} \leq \sqrt{\frac{2\sigma^2 I(U, W_{1:n}; S_{1:n}^{va}|S_{1:n}^{tr})}{nm_{va}}}.$$

*Proof.* The proof technique is analogous to Theorem 5.1. Let $\Phi = (U, W_{1:n})$ be a collection of random variables where $\Phi \in \mathcal{U} \times \mathcal{W}^n$ such that $\Phi$ and $S_{1:n}$ follow the joint distribution $P_{\Phi, S_{1:n}}$. Then let $\tilde{\Phi}$ be an independent copy of $\Phi$, such that $\tilde{\Phi} \perp\!\!\!\perp \{S_{1:n}^{va}, S_{1:n}^{tr}\}$, i.e., $\tilde{\Phi} \sim \mathbb{E}_{S_{1:n}} P_{\Phi|S_{1:n}}$. Define

$$f(\Phi, S_{1:n}^{va}) = \frac{1}{n} \sum_{i=1}^{n} [R_{S_i^{va}}(W_i)] = \frac{1}{n} \sum_{i=1}^{n} \frac{1}{m} \sum_{j=1}^{m_{va}} \ell(W_i, Z_{i,j}).$$

For any $\lambda \in \mathbb{R}$, denote the cumulant generation function of $\tilde{\Phi}, S_{1:n}^{va}|S_{1:n}^{tr}$ as:

$$\psi_{\tilde{\Phi}, S_{1:n}^{va}|S_{1:n}^{tr}}(\lambda) = \log \mathbb{E}_{\tilde{\Phi}, S_{1:n}^{va}|S_{1:n}^{tr}} [e^{\lambda(f(\tilde{\Phi}, S_{1:n}^{va}) - \mathbb{E}_{\tilde{\Phi}, S_{1:n}^{va}|S_{1:n}^{tr}}[f(\tilde{\Phi}, S_{1:n}^{va})]})]$$

$$= \log \mathbb{E}_{\tilde{\Phi}, S_{1:n}^{va}|S_{1:n}^{tr}} [e^{\lambda f(\tilde{\Phi}, S_{1:n}^{va})}] - \lambda \mathbb{E}_{\tilde{\Phi}, S_{1:n}^{va}|S_{1:n}^{tr}} [f(\tilde{\Phi}, S_{1:n}^{va})]$$

In addition, the disintegrated mutual information is given as:

$$I^{S^{\text{tr}}_{1:n}}(\Phi; S^{\text{va}}_{1:n}) = D_{\text{KL}}\big(P_{\Phi, S^{\text{va}}_{1:n}|S^{\text{tr}}_{1:n}} || P_{\tilde{\Phi}|S^{\text{tr}}_{1:n}} P_{S^{\text{va}}_{1:n}|S^{\text{tr}}_{1:n}}\big)$$

$$= \sup_{g} \left\{ \mathbb{E}_{\Phi, S^{\text{va}}_{1:n}|S^{\text{tr}}_{1:n}}[g(\Phi, S^{\text{va}}_{1:n})] - \log \mathbb{E}_{\tilde{\Phi}, S^{\text{va}}_{1:n}|S^{\text{tr}}_{1:n}}\left[e^{g(\tilde{\Phi}, S^{\text{va}}_{1:n})}\right]\right\}$$

$$\geq \lambda \mathbb{E}_{\Phi, S^{\text{va}}_{1:n}|S^{\text{tr}}_{1:n}}[f(\Phi, S^{\text{va}}_{1:n})] - \log \mathbb{E}_{\tilde{\Phi}, S^{\text{va}}_{1:n}|S^{\text{tr}}_{1:n}}\left[e^{\lambda f(\tilde{\Phi}, S^{\text{va}}_{1:n})}\right]$$

$$= \lambda \mathbb{E}_{U, W_{1:n}, S^{\text{va}}_{1:n}|S^{\text{tr}}_{1:n}}[f(\Phi, S^{\text{va}}_{1:n})]$$

$$- \lambda \mathbb{E}_{\tilde{U}, \tilde{W}_{1:n}, S^{\text{va}}_{1:n}|S^{\text{tr}}_{1:n}}[f(\tilde{\Phi}, S^{\text{va}}_{1:n})] - \psi_{\tilde{\Phi}, S^{\text{va}}_{1:n}|S^{\text{tr}}_{1:n}}(\lambda)$$

$$= \lambda \mathbb{E}_{U, W_{1:n}, S^{\text{va}}_{1:n}|S^{\text{tr}}_{1:n}} \frac{1}{n}\sum_{i=1}^{n} R_{S^{\text{va}}_i}(W_i)$$

$$- \lambda \mathbb{E}_{\tilde{U}, \tilde{W}_{1:n}, S^{\text{va}}_{1:n}|S^{\text{tr}}_{1:n}} \frac{1}{n}\sum_{i=1}^{n} R_{S^{\text{va}}_i}(\tilde{W}_i) - \psi_{\tilde{\Phi}, S^{\text{va}}_{1:n}|S^{\text{tr}}_{1:n}}(\lambda) \quad (7)$$

Since given $U$, $(W_i, S^{\text{tr}}_i), i = 1, ..., n$ are mutually independent, we have $p(w_{1:n}|s^{\text{tr}}_{1:n}, u) = \prod_{i=1}^{n} p(w_i|s^{\text{tr}}_i, u)$. Thus

$$\lambda \mathbb{E}_{U, W_{1:n}, S^{\text{va}}_{1:n}|S^{\text{tr}}_{1:n}} \frac{1}{n}\sum_{i=1}^{n}[R_{S^{\text{va}}_i}(W_i)] = \lambda \mathbb{E}_{U, S^{\text{va}}_{1:n}|S^{\text{tr}}_{1:n}} \frac{1}{n}\sum_{i=1}^{n} \mathbb{E}_{W_i|S^{\text{tr}}_i, U}[R_{S^{\text{va}}_i}(W_i)]$$

$$= \lambda \mathbb{E}_{U, S^{\text{va}}_{1:n}|S^{\text{tr}}_{1:n}}[\tilde{R}_{S_{1:n}}(U)] \quad (8)$$

Since we have $\tilde{\Phi} \perp\!\!\!\perp \{S^{\text{va}}_{1:n}, S^{\text{tr}}_{1:n}\}$, thus $P_{\tilde{W}_{1:n}|S_{1:n}, \tilde{U}} = P_{\tilde{W}_{1:n}|\tilde{U}}$, we have:

$$R_\tau(\tilde{U}) = \mathbb{E}_{S \sim \mu_{m,\tau}} \mathbb{E}_{\tilde{W} \sim P_{\tilde{W}|S, \tilde{U}}}[R_\mu(\tilde{W})] = \mathbb{E}_{\mu \sim \tau} \mathbb{E}_{S|\mu \sim \mu^m} \mathbb{E}_{\tilde{W} \sim P_{\tilde{W}|\tilde{U}}}[R_\mu(\tilde{W})]$$

$$= \mathbb{E}_{\mu \sim \tau} \mathbb{E}_{\tilde{W} \sim P_{\tilde{W}|\tilde{U}}}[R_\mu(\tilde{W})]$$

Moreover, we have $S^{\text{tr}}_{1:n} \perp\!\!\!\perp S^{\text{va}}_{1:n}$, so that $P_{\tilde{W}_{1:n}, S^{\text{va}}_{1:n}, \tilde{U}|S^{\text{tr}}_{1:n}} = P_{\tilde{W}_{1:n}, \tilde{U}} P_{S^{\text{va}}_{1:n}|S^{\text{tr}}_{1:n}} = P_{\tilde{W}_{1:n}, \tilde{U}} P_{S^{\text{va}}_{1:n}}$. Then we can also prove:

$$\lambda \mathbb{E}_{\tilde{U}, \tilde{W}_{1:n}, S^{\text{va}}_{1:n}|S^{\text{tr}}_{1:n}}\left[\frac{1}{n}\sum_{i=1}^{n} R_{S^{\text{va}}_i}(\tilde{W}_i)\right] = \lambda \mathbb{E}_{\tilde{U}, \tilde{W}_{1:n}|S^{\text{tr}}_{1:n}} \mathbb{E}_{S^{\text{va}}_{1:n} \sim \mu^n_{m_{\text{va}}, \tau}}\left[\frac{1}{n}\sum_{i=1}^{n} R_{S^{\text{va}}_i}(\tilde{W}_i)\right]$$

$$= \lambda \mathbb{E}_{\tilde{U}, \tilde{W}_{1:n}|S^{\text{tr}}_{1:n}}\left[\frac{1}{n}\sum_{i=1}^{n} \mathbb{E}_{S_i \sim \mu_{m_{\text{va}}, \tau}} R_{S^{\text{va}}_i}(\tilde{W}_i)\right]$$

$$= \lambda \mathbb{E}_{\tilde{U}, \tilde{W}_{1:n}|S^{\text{tr}}_{1:n}}\left[\frac{1}{n}\sum_{i=1}^{n} \mathbb{E}_{\mu_i \sim \tau} \mathbb{E}_{S_i|\mu_i \sim \mu^{m_{\text{va}}}_i}[R_{S^{\text{va}}_i}(\tilde{W}_i)]\right]$$

$$= \lambda \mathbb{E}_{\tilde{U}, \tilde{W}_{1:n}|S^{\text{tr}}_{1:n}}\left[\frac{1}{n}\sum_{i=1}^{n} \mathbb{E}_{\mu_i \sim \tau}\left[\frac{1}{m_{\text{va}}}\sum_{j=1}^{m_{\text{va}}} \mathbb{E}_{Z_{i,j} \sim \mu_i} \ell(\tilde{W}_i, Z_{i,j})\right]\right]$$

$$= \lambda \mathbb{E}_{\tilde{U}|S^{\text{tr}}_{1:n}}\left[\frac{1}{n}\sum_{i=1}^{n} \mathbb{E}_{\tilde{W}_i|\tilde{U}} \mathbb{E}_{\mu_i \sim \tau}\left[\frac{1}{m_{\text{va}}}\sum_{j=1}^{m_{\text{va}}} \mathbb{E}_{Z_{i,j} \sim \mu_i} \ell(\tilde{W}_i, Z_{i,j})\right]\right]$$

$$= \lambda \mathbb{E}_{\tilde{U}|S^{\text{tr}}_{1:n}}\left[\mathbb{E}_{\mu \sim \tau} \mathbb{E}_{\tilde{W}|\tilde{U}}[R_\mu(\tilde{W})]\right]$$

$$= \lambda \mathbb{E}_{\tilde{U}|S^{\text{tr}}_{1:n}} R_\tau(\tilde{U}) = \lambda \mathbb{E}_{U, S^{\text{va}}_{1:n}|S^{\text{tr}}_{1:n}} R_\tau(U) \quad (9)$$

Therefore, by combining Equations (7), (8), and (9), we have for any $\lambda$,

$$\lambda \mathbb{E}_{U,S^{\text{va}}_{1:n}|S^{\text{tr}}_{1:n}}[\tilde{R}_{S_{1:n}}(U)] - \lambda \mathbb{E}_{U,S^{\text{va}}_{1:n}|S^{\text{tr}}_{1:n}}[R_\tau(U)] \le I^{S^{\text{tr}}_{1:n}}(\Phi; S^{\text{va}}_{1:n}) + \psi_{\tilde{\Phi}, S^{\text{va}}_{1:n}|S^{\text{tr}}_{1:n}}(\lambda)$$

Since $\ell(\tilde{W}, Z)$ is $\sigma$-subgaussian, and $\tilde{\Phi} \perp\!\!\!\perp S^{\text{va}}_{1:n}$, $f(\tilde{\Phi}, S^{\text{va}}_{1:n}) = \frac{1}{n}\sum_{i=1}^{n}\frac{1}{m_{\text{va}}}\sum_{j=1}^{m_{\text{va}}}\ell(\tilde{W}_i, Z_{ij})$ is $\frac{\sigma}{\sqrt{nm_{\text{va}}}}$-subgaussian. Hence, $\psi_{\tilde{\Phi}, S^{\text{va}}_{1:n}|S^{\text{tr}}_{1:n}}(\lambda) \le \frac{\lambda^2\sigma^2}{2nm_{\text{va}}}, \forall \lambda \in \mathbb{R}$. For $\lambda < 0$ we have

$$\mathbb{E}_{U,S^{\text{va}}_{1:n}|S^{\text{tr}}_{1:n}}[R_\tau(U) - \tilde{R}_{S_{1:n}}(U)] \le \frac{I^{S^{\text{tr}}_{1:n}}(\Phi; S^{\text{va}}_{1:n}) + \psi_{\tilde{\Phi}, S^{\text{va}}_{1:n}|S^{\text{tr}}_{1:n}}(\lambda)}{-\lambda} \le \sqrt{\frac{2\sigma^2 I^{S^{\text{tr}}_{1:n}}(\Phi; S^{\text{va}}_{1:n})}{nm_{\text{va}}}}$$

Similarly, for $\lambda > 0$ we have

$$\mathbb{E}_{U,S^{\text{va}}_{1:n}|S^{\text{tr}}_{1:n}}[R_\tau(U) - \tilde{R}_{S_{1:n}}(U)] \ge \frac{I^{S^{\text{tr}}_{1:n}}(\Phi; S^{\text{va}}_{1:n}) + \psi_{\tilde{\Phi}, S^{\text{va}}_{1:n}|S^{\text{tr}}_{1:n}}(\lambda)}{-\lambda} \ge -\sqrt{\frac{2\sigma^2 I^{S^{\text{tr}}_{1:n}}(\Phi; S^{\text{va}}_{1:n})}{nm_{\text{va}}}}$$

Then, the following concludes the proof:

$$|\text{gen}^{\text{alt}}_{\text{meta}}(\tau, \mathcal{A}_{\text{meta}}, \mathcal{A}_{\text{base}})| = \mathbb{E}_{S^{\text{tr}}_{1:n}}|\mathbb{E}_{U,S^{\text{va}}_{1:n}|S^{\text{tr}}_{1:n}}[R_\tau(U) - \tilde{R}_{S_{1:n}}(U)]| \le \mathbb{E}_{S^{\text{tr}}_{1:n}}\sqrt{\frac{2\sigma^2 I^{S^{\text{tr}}_{1:n}}(\Phi; S^{\text{va}}_{1:n})}{nm_{\text{va}}}}$$

$\square$

## B.4 Proof of Theorem 6.1

**Theorem.** *Based on Theorem 5.1, for the SGLD algorithm that satisfies Assumptions 1 & 2, the mutual information for joint training satisfies*

$$I(\Phi; S_{1:n}) \le \sum_{t=1}^{T}\frac{nd+k}{2}\log\left(1 + \frac{\eta_t^2 L^2}{(nd+k)\sigma_t^2}\right).$$

*Specifically, if $\sigma_t = \sqrt{\eta_t}$, and $\eta_t = \frac{c}{t}$ for $c > 0$, we have:*

$$|gen^{joi}_{meta}(\tau, \mathcal{A}_{meta}, \mathcal{A}_{base})| \le \frac{\sigma L}{\sqrt{nm}}\sqrt{c\log T + c}.$$

*Proof.* Define the sequence of parameters for $T$ iterations as $\Phi^{[T]} \overset{\text{def}}{=} (\Phi^1, ..., \Phi^T)$ and the corresponding sequence of samplings as $B^{[T]}_{1:n} \overset{\text{def}}{=} (B^1_{1:n}, ..., B^T_{1:n})$. The output of the algorithm is defined as $\Phi = f(\Phi^{[T]})$, which can be the last iterate $\Phi^T$ or the average output $\frac{1}{T}\sum_{t=1}^{T}\Phi^t$. From the figure about the parameter updating strategy for joint training illustrated in Section 6, we get the following Markov chain:

$$S_{1:n} \to B^{[T]}_{1:n} \to \Phi^{[T]} \to \Phi.$$

Therefore, by applying Lemma A.7 to the above Markov chain, we have:

$$I(\Phi; S_{1:n}) \le I(\Phi^{[T]}; S_{1:n}) \le I(\Phi^{[T]}; B^{[T]}_{1:n}) = \sum_{t=1}^{T}I(\Phi^t; B^{[T]}_{1:n}|\Phi^{[t-1]}).$$

The last equality comes from the mutual information chain rule. Combing the sample strategy with Assumption 1 and the update rule, we obtain:

$$I(\Phi^t; B^{[T]}_{1:n}|\Phi^{[t-1]}) = I(\Phi^t; B^t_{1:n}|\Phi^{t-1})$$
$$= h(\Phi^t|\Phi^{t-1}) - h(\Phi^t|\Phi^{t-1}, B^t_{1:n}).$$

Conditioned on $\Phi^{t-1} = \phi^{t-1}$, we have $\Phi^t = \phi^{t-1} - \eta_t G(\phi^{t-1}, B^t_{1:n}) + \xi^t$. Then

$$h(\Phi^t - \phi^{t-1}|\Phi^{t-1} = \phi^{t-1}) = h(\Phi^t|\Phi^{t-1} = \phi^{t-1}).$$

Note that $\xi^t$ and $\eta_t G(\phi^{t-1}, B^t_{1:n})$ are independent. So we have

$$\mathbb{E}(||\Phi^t - \phi^{t-1}||_2^2) = \mathbb{E}(||\eta_t G(\phi^{t-1}, B^t_{1:n})||_2^2 + ||\xi^t||_2^2) \le \eta_t^2 L^2 + (nd+k)\sigma_t^2.$$

The Gaussian distribution is the one having the largest entropy among the variables with the same second order moment. Hence,

$$h(\Phi^t|\Phi^{t-1} = \phi^{t-1}) \le \frac{nd+k}{2}\log(2\pi e\frac{\eta_t^2 L^2 + (nd+k)\sigma_t^2}{(nd+k)})$$

for all $\phi^{t-1}$.

In addition,

$$h(\Phi^t|\Phi^{t-1}, B_{1:n}^t) = h(\Phi^{t-1} - \eta_t G(\Phi^{t-1}, B_{1:n}^t) + \xi^t|\Phi^{t-1}, B_{1:n}^t)$$
$$= h(\xi^t) = \frac{nd+k}{2}\log 2\pi e\sigma_t^2.$$

So we obtain

$$I(\Phi; S_{1:n}) \le \sum_{t=1}^{T} \frac{nd+k}{2}\log(1 + \frac{\eta_t^2 L^2}{(nd+k)\sigma_t^2}) \le \sum_{t=1}^{T} \frac{\eta_t^2 L^2}{2\sigma_t^2}.$$

Hence, for the SGLD algorithm with $\sigma_t = \sqrt{\eta_t}$, constant $c > 0$, $\eta_t = \frac{c}{t}$; since $\sum_{t=1}^{T} \frac{1}{t} \le \log T + 1$, we have

$$|\text{gen}_{\text{meta}}^{\text{joi}}(\tau, \mathcal{A}_{\text{meta}}, \mathcal{A}_{\text{base}})| \le \sqrt{\frac{2\sigma^2(I(U, W_{1:n}; S_{1:n}))}{nm}}$$
$$\le \sqrt{\frac{\sigma^2}{nm}\sum_{t=1}^{T} \frac{\eta_t^2 L^2}{2\sigma_t^2}}$$
$$\le \frac{\sigma L}{\sqrt{nm}}\sqrt{c\log T + c}.$$

$\square$

## B.5 Proof of Theorem 6.2

**Theorem.** *Based on Theorem 5.2, for the Meta-SGLD that satisfies Assumption 1, if we set $\sigma_t = \sqrt{2\eta_t/\gamma_t}$, $\sigma_{t,k} = \sqrt{2\beta_{t,k}/\gamma_{t,k}}$, where $\gamma_t$ and $\gamma_{t,k}$ are the inverse temperatures. The meta generalization error for alternate training satisfies*

$$|gen_{meta}^{alt}(\tau, SGLD, SGLD)| \le \sqrt{\frac{2\sigma^2 I(U, W_{1:n}; S_{1:n}^{va}|S_{1:n}^{tr})}{nm_{va}}} \le \frac{\sigma}{\sqrt{nm_{va}}}\sqrt{\epsilon_U + \epsilon_W},$$

*where*

$$\epsilon_U = \sum_{t=1}^{T} \mathbb{E}_{B_{I_t}^{va}, B_{I_t}^{tr}, W_{I_t}, U^{t-1}} \frac{\eta_t\gamma_t\|\epsilon_t^u\|_2^2}{2}, \quad \epsilon_W = \sum_{t=1}^{T}\sum_{i=1}^{|I_t|}\sum_{k=1}^{K} \mathbb{E}_{B_{i,t,k}^{va}, B_{i,t,k}^{tr}, W_{i,t}^{k-1}} \frac{\beta_{t,k}\gamma_{t,k}\|\epsilon_{t,i,k}^w\|_2^2}{2}.$$

*Proof.* To prove the above theorem, we need to introduce some basic notations to present the sampling results and the intermediate output of each gradient step, by which we can apply the Markov structure and the mutual information chain rule.

- for $K$ inner iterations:
    - The sequence of validation data samplings at outer iteration $t$ for task $i$ and the task batch:
    $$B_{i,t,[K]}^{va} = (B_{i,t,1}^{va}, ..., B_{i,t,K}^{va}), B_{I_t,[K]}^{va} = (B_{1,t,[K]}^{va}, ..., B_{|I_t|,t,[K]}^{va})$$
    - The sequence of train data samplings at outer iteration $t$ for task $i$ and the task batch:
    $$B_{i,t,[K]}^{tr} = (B_{i,t,1}^{tr}, ..., B_{i,t,K}^{tr}), B_{I_t,[K]}^{tr} = (B_{1,t,[K]}^{tr}, ..., B_{|I_t|,t,[K]}^{tr})$$

- the sequence of task specific parameters at outer iteration $t$ of task $i$ and the task batch:
$$W_{i,t}^{[K]} = (W_{i,t}^1, ..., W_{i,t}^K), W_{I_t}^{[K]} = (W_{1,t}^{[K]}, ..., W_{|I_t|,t}^{[K]});$$
- The output of base learner at outer iteration $t$ of task $i$ and the task batch:
$$W_{i,t} = g(W_{i,t}^{[K]}), W_{I_t} = (W_{1,t}, ..., W_{|I_t|,t})$$

- for $T$ outer iterations:

  - The sequence of meta parameters as $U^{[T]} = (U^1, ..., U^T)$;
  - validation data sequences as $B_{I_{[T]}}^{\text{va}} = (B_{I_1}^{\text{va}}, ..., B_{I_T}^{\text{va}})$;
  - train data sequences as $B_{I_{[T]}}^{\text{tr}} = (B_{I_1}^{\text{tr}}, ..., B_{I_T}^{\text{tr}})$;
  - Output of meta learner is defined as $U = f(U^{[T]})$;
  - Output sequence of base learner is defines as $W_{I_{[T]}} = (W_{I_1}, ..., W_{I_T})$

Based on the definition above, we have the following Markov chains:
$$S_{1:n}^{\text{va}} \to B_{I_{[T]}}^{\text{va}} \to U^{[T]} \to U \tag{10}$$
$$S_{1:n}^{\text{tr}} \to B_{I_{[T]}}^{\text{tr}} \to W_{I_{[T]}} \to W_{1:n} \tag{11}$$
$$B_{I_t}^{\text{tr}} \to B_{I_t,[K]}^{\text{tr}} \to W_{I_t}^{[K]} \to W_{I_t} \tag{12}$$
$$B_{i,t}^{\text{tr}} \to B_{i,t,[K]}^{\text{tr}} \to W_{i,t}^{[K]} \to W_{i,t} \tag{13}$$

And the graph model:
$$S_{1:n}^{\text{tr}} \to B_{I[T]}^{\text{tr}} \to (W_{I_{[T]}}, U^{[T]}) \leftarrow S_{1:n}^{\text{va}} \tag{14}$$

In fact, the algorithm has a nest-loop structure, we just list the above simple sub-structures for the first step of the proof. By combining the above Markov chains and the independence of the sample strategy, we obtain

$$I(U, W_{1:n}; S_{1:n}^{\text{va}} | S_{1:n}^{\text{tr}}) \leq I(U^{[T]}, W_{1:n}; S_{1:n}^{\text{va}} | S_{1:n}^{\text{tr}}) \leq I(U^{[T]}, W_{I_{[T]}}; S_{1:n}^{\text{va}} | S_{1:n}^{\text{tr}})$$
$$\leq I(U^{[T]}, W_{I_{[T]}}; S_{1:n}^{\text{va}} | B_{I_{[T]}}^{\text{tr}}) \leq I(U^{[T]}, W_{I_{[T]}}; B_{I_{[T]}}^{\text{va}} | B_{I_{[T]}}^{\text{tr}}) \tag{15}$$

Apply Lemma A.7, the first and the last inequality are obtained with Markov chain (10). The second inequality is obtained with (11). The third inequality comes from Lemma A.8 and the graph model(14).

Furthermore, we can apply (12), (13), the information chain rule together with the updating rules, to obtain the following decomposition:

$$I(U^{[T]}, W_{I_{[T]}}; B_{I_{[T]}}^{\text{va}} | B_{I_{[T]}}^{\text{tr}}) = \sum_{t=1}^{T} I(U^t, W_{I_t}; B_{I_t}^{\text{va}} | B_{I_t}^{\text{tr}}, U^{t-1}, W_{I_{t-1}})$$

$$= \sum_{t=1}^{T} \left\{ I(W_{I_t}; B_{I_t}^{\text{va}} | B_{I_t}^{\text{tr}}, U^{t-1}) + I(U^t; B_{I_t}^{\text{va}} | B_{I_t}^{\text{tr}}, W_{I_t}, U^{t-1}) \right\}$$

$$\leq \sum_{t=1}^{T} \left\{ \sum_{i=1}^{b} \left[ I(W_{i,t}^{[K]}; B_{i,t}^{\text{va}} | B_{i,t}^{\text{tr}}, U^{t-1}) \right] + I(U^t; B_{I_t}^{\text{va}} | B_{I_t}^{\text{tr}}, U^{t-1}, W_{I_t}) \right\}$$

$$\leq \sum_{t=1}^{T} \sum_{i=1}^{b} \sum_{k=1}^{K} I(W_{i,t}^k; B_{i,t,k}^{\text{va}} | U^{t-1}, B_{i,t,k}^{\text{tr}}, W_{i,t}^{k-1}) + \sum_{t=1}^{T} I(U^t; B_{I_t}^{\text{va}} | B_{I_t}^{\text{tr}}, U^{t-1}, W_{I_t})$$

$$= \sum_{t=1}^{T} \sum_{i=1}^{b} \sum_{k=1}^{K} \mathbb{E}_{B_{i,t,k}^{\text{va}}, B_{i,t,k}^{\text{tr}}, W_{i,t}^{k-1}} \left[ D_{\text{KL}}(P_{W_{i,t}^k | B_{i,t,k}^{\text{tr}}, B_{i,t,k}^{\text{va}}, W_{i,t}^{k-1}} || P_{W_{i,t}^k | B_{i,t,k}^{\text{tr}}, W_{i,t}^{k-1}}) \right]$$

$$+ \sum_{t=1}^{T} \mathbb{E}_{B_{I_t}^{\text{va}}, B_{I_t}^{\text{tr}}, U^{t-1}} \left[ D_{\text{KL}}(P_{U^t | B_{I_t}^{\text{va}}, B_{I_t}^{\text{tr}}, U^{t-1}, W_{I_t}} || P_{U^t | B_{I_t}^{\text{tr}}, U^{t-1}, W_{I_t}}) \right]. \tag{16}$$

**Remark.** *Here, the KL divergence is for every single iteration, it's not for the full trajectory. In addition, the randomness brought by sampling and previous updates is implied by the expectation. To empirically evaluate the bound, we can sample the variables presented in the expectation to compute the KL divergence.*

Recall the following updates rules:

$$W_{i,t}^k = W_{i,t}^{k-1} - \beta_{t,k}\nabla R_{B_{i,t,k}^{\text{tr}}}(W_{i,t}^{k-1}) + \zeta^{t,k}$$
$$U^t = U^{t-1} - \eta_t\nabla\tilde{R}_{B_{I_t}^{\text{va}}}(U^{t-1}) + \xi^t.$$

For the SGLD algorithm, we use the typical choices of $\sigma_t = \sqrt{2\eta_t/\gamma_t}$, $\zeta_k = \sqrt{2\beta_{t,k}/\gamma_{t,k}}$, where $\gamma_t$ and $\gamma_{t,k}$ are the inverse temperatures. Then, the update rules give

$$P_{U^t|B_{I_t}^{\text{tr}},U^{t-1},W_{I_t}} \sim \mathcal{N}(U^{t-1} - \eta_t\nabla\tilde{R}_{B_{I_t}^{\text{tr}}}(U^{t-1}), \frac{2\eta_t}{\gamma_t})$$

$$P_{U^t|B_{I_t}^{\text{va}},B_{I_t}^{\text{tr}},U^{t-1},W_{I_t}} \sim \mathcal{N}(U^{t-1} - \eta_t\nabla\tilde{R}_{B_{I_t}^{\text{va}},B_{I_t}^{\text{tr}}}(U^{t-1}), \frac{2\eta_t}{\gamma_t})$$

$$P_{W_{i,t}^k|B_{i,t,k}^{\text{tr}},W_{i,t}^{k-1}} \sim \mathcal{N}(W_{i,t}^{k-1} - \beta_{t,k}\nabla R_{B_{i,t,k}^{\text{tr}}}(W_{i,t}^{k-1}), \frac{2\beta_{t,k}}{\gamma_{t,k}})$$

$$P_{W_{i,t}^k|B_{i,t,k}^{\text{tr}},B_{i,t,k}^{\text{va}},W_{i,t}^{k-1}} \sim \mathcal{N}(W_{i,t}^{k-1} - \beta_{t,k}\nabla R_{B_{i,t,k}^{\text{tr}},B_{i,t,k}^{\text{va}}}(W_{i,t}^{k-1}), \frac{2\beta_{t,k}}{\gamma_{t,k}})$$

Let $\epsilon_t^u = \nabla\tilde{R}_{B_{I_t}^{\text{va}},B_{I_t}^{\text{tr}}}(U^{t-1}) - \nabla\tilde{R}_{B_{I_t}^{\text{tr}}}(U^{t-1})$, then we have

$$D_{\text{KL}}(P_{U^t|B_{I_t}^{\text{va}},B_{I_t}^{\text{tr}},W_{I_t}}||P_{U^t|B_{I_t}^{\text{tr}},W_{I_t}}) = \frac{\eta_t^2||\epsilon_t^u||_2^2}{2\sigma_t^2} = \frac{\eta_t\gamma_t||\epsilon_t^u||_2^2}{4} \tag{17}$$

Similarly, let $\epsilon_{t,i,k}^w = \nabla R_{\tilde{B}_{i,t,k}^{\text{tr}},\tilde{B}_{i,t,k}^{\text{va}}}(W_{i,t}^{k-1}) - \nabla R_{\tilde{B}_{i,t,k}^{\text{tr}}}(W_{i,t}^{k-1})$, we have

$$D_{\text{KL}}(P_{W_{i,t}^k|W_{i,t}^{k-1},B_{i,t,k}^{\text{va}},B_{i,t,k}^{\text{tr}}}||P_{W_{i,t}^k|W_{i,t}^{k-1},B_{i,t,k}^{\text{tr}}}) = \frac{\beta_{t,k}\gamma_{t,k}||\epsilon_{t,i,k}^w||_2^2}{4} \tag{18}$$

Combine Theorem 5.2 and equations(15), (16), (17),(18), we have

$$|\text{gen}_{\text{meta}}^{\text{alt}}(\tau,\text{SGLD},\text{SGLD})| \leq \sqrt{\frac{2\sigma^2(I(U,W_{1:n};S_{1:n}^{\text{va}}|S_{1:n}^{\text{tr}}))}{nm_{\text{va}}}}$$

$$\leq \frac{\sigma}{\sqrt{nm_{\text{va}}}}\sqrt{\sum_{t=1}^T\mathbb{E}_{B_{I_t}^{\text{va}},B_{I_t}^{\text{tr}},U^{t-1},W_{I_t}}\frac{\eta_t\gamma_t||\epsilon_t^u||_2^2}{2} + \sum_{t=1}^T\sum_{i=1}^{|I_t|}\sum_{k=1}^K\mathbb{E}_{B_{i,t,k}^{\text{va}},B_{i,t,k}^{\text{tr}},W_{i,t}^{k-1}}\frac{\beta_t^k\gamma_t^k||\epsilon_{t,i,k}^w||_2^2}{2}}$$

which concludes the proof. $\square$

## C  On Subgaussianity

We list the two subgaussian assumptions of Xu and Raginsky [15] and Bu et al. [37] respectively as follows:

**Assumption (a)** $\forall w \in \mathcal{W}$, $\ell(w,Z)$ is $\sigma$-subgaussian for $Z \sim \mu$.

**Assumption (b)** $\ell(\tilde{W},Z)$ is $\sigma$-subgaussian under $P_{\tilde{W},Z} = P_W \times \mu$, where $\tilde{W}$ is an independent copy of $W$ and $\tilde{W} \perp\!\!\!\perp Z$.

Xu and Raginsky [15] directly use Assumption (a) to conclude Assumption (b) in their proof. Two counter examples have been proposed to challenge this conclusion in Appendix section C of [39] and section IV of [37]. However, we notice that these two counterexamples are based on the case of unbounded loss with no constraint on the parameter $W$ output by the learning algorithm. We now compare the two assumptions mentioned above in detail for unbounded loss and bounded loss.

## C.1 unbounded loss

**Counterexample for Assumption (a) => (b)** (Negrea et al. [39])

Consider $\mathcal{W} = \mathcal{Z} = \mathbb{R}$ with $\ell(w, z) = w + z$. Assume that $\tilde{W} \perp\!\!\!\perp Z, \tilde{W} \sim Cauchy, \tilde{Z} \sim \mathcal{N}(0, \sigma^2)$. Thus, $\ell(w, Z)$ is $\sigma$-subgaussian for any $w \in \mathcal{W}$, because $\psi_{\ell(w,Z)}(\lambda) = \log \mathbb{E}_Z[e^{\lambda(\ell(w,Z) - \mathbb{E}\ell(w,Z))}] = \log \mathbb{E}[e^{\lambda Z}] = \exp \frac{\lambda^2 \sigma^2}{2}$. While $\ell(\tilde{W}, \tilde{Z})$ is not subgaussian since the Cauchy distribution does not have well-defined moments higher than the zeroth moment.

**Counterexample for Assumption (b) => (a)** (Bu et al. [37])

Consider $\mathcal{W} = \mathcal{Z} = \mathbb{R}^d$ and the square loss function $\ell(w, z) = \|w - z\|_2^2$. Assume $\tilde{W} \sim \mathcal{N}(\mu, \sigma_W^2 \mathbb{I}_d), Z \sim \mathcal{N}(\mu, \sigma_Z^2 \mathbb{I}_d)$. Then $\ell(w, Z)$ is not subgaussian for all $w \in \mathcal{W}$, since when $\|w\|_2^2 \to \infty$ the variance of $\ell(w, Z)$ is not bounded. However, $\tilde{W} - Z \sim \mathcal{N}(0, (\sigma_W^2 + \sigma_Z^2)\mathbb{I}_d)$, so $\ell(\tilde{W}, \tilde{Z}) = \|\tilde{W} - Z\|_2^2 \sim (\sigma_W^2 + \sigma_Z^2)\chi_d^2$ has bounded CGF for $\lambda < 0$, which can induce one-sided bound in our theorem with $\sigma^2 = 2d(\sigma_W^2 + \sigma_Z^2)^2$. However, in this condition, the loss is sub-exponential but not subgaussian as claimed by Bu et al. [37] in assumption (b).

## C.2 bounded loss

For a bounded loss function $\ell(w, z) \in [a, b]$, the two assumptions are equivalent. $\ell(w, Z)$ is $\frac{(b-a)}{2}$-subgaussian $\forall w \in \mathcal{W}, Z \sim \mu$. Similarly, $\ell(W, Z)$ is $\frac{(b-a)}{2}$-subgaussian under $P_{\tilde{W}, Z} = P_W \times \mu$. The counter example of [39] does not apply because the Cauchy distribution is truncated and, consequently, has well-defined moments.

## C.3 Discussion

Based on the above analysis, we can conclude the following. For a bounded loss, the two assumptions are equivalent. In contrast, Assumption (b) is a stronger assumption than Assumption (a) when the loss function is unbounded. At the same time, we found that Assumption (b) is also hard to ensure and is often replaced by the sub-exponential assumption as a relaxation for unbounded loss.

What we need for proving Theorem 5.1 and 5.2 is actually the extension of assumption (b). However, in practice, the parameters output from an algorithm should always be bounded. Moreover, for complex data sets used in deep learning, people often adopt a bounded loss or truncate the unbounded loss to ensure the theoretical guarantee. The inconsistency between the two assumptions should not cause too many problems. Hence, we extended Assumption (a) to avoid confusion and too much discussion in the main paper, although the more rigorous version should make use of Assumption (b).

# D   Additional Experimental Results

## D.1   Synthetic Data

In this section, we present a more direct visualization for the 2D mean estimation experiment described in Section 7.1. We compare the results of three different train-validation split settings in Figure 4. The yellow cross in the figure is the actual environment mean $(-4, -4)$. Note that we have set the task batch size as $|I_t| = 5$. The five clusters in the graph are the task batch data points at the last epoch, which corresponds to five different $\mu_i \sim \tau, \forall i \in [|I_t|]$. We use small dots to represent the data points, and big dots to show the estimated cluster mean $W_i$ and the estimated environment mean $U$.

Figure 4 illustrates that the distances from the estimated $U$ to the yellow cross are slightly different for these three settings. When $m_{\mathrm{va}} = 1$ the estimated mean $U$ is much closer to the actual environment mean. This result is coherent with the bound estimation results in Section 7.1, where we got the tightest gradient incoherence bound with $m_{\mathrm{va}} = 1$. While the gradient norm bound is largest for $m_{\mathrm{va}} = 1$, which indicates that the gradient norm bound may not be as reliable as the gradient incoherence bound since it may be much looser and won't give too much information.

**Comparison with the observed generalization error**

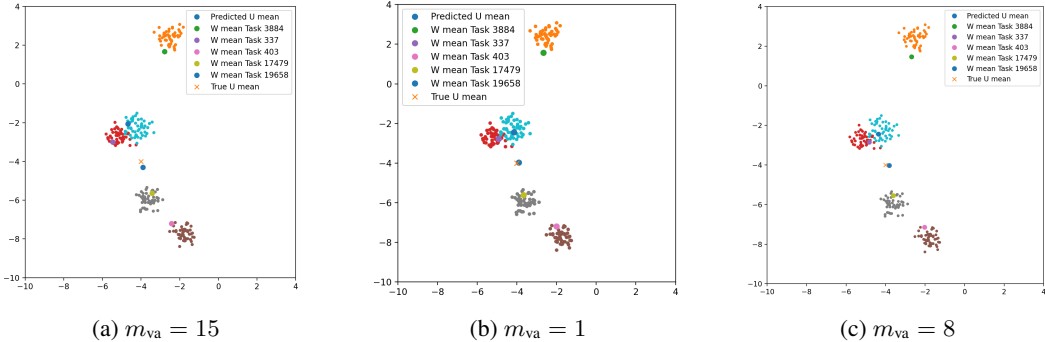

|              |              |              |
|:------------:|:------------:|:------------:|
| (a) $m_{va} = 15$ | (b) $m_{va} = 1$ | (c) $m_{va} = 8$ |

Figure 4: Visualization for simulated data results

We calculated the observed generalization error by evaluating the expected difference between the train loss and test loss. And we list the results of synthetic data under different train-validation split settings in Table 1, Table 2 and Table 3.

Table 1: $m_{tr} = 8, m_{va} = 8$

| epoch | 20 | 40 | 60 | 80 | 100 | 120 | 140 | 160 | 180 |
|---|---|---|---|---|---|---|---|---|---|
| *Train-Test gap* | 0.0697 | 0.0371 | 0.0908 | 0.0241 | 0.1072 | 0.1492 | 0.1775 | 0.1432 | 0.1581 |
| *Lipschitz* | 19.77 | 27.63 | 33.7 | 38.84 | 43.37 | 47.47 | 51.24 | 54.75 | 58.05 |
| *G_norm* | 6.009 | 7.506 | 8.807 | 9.856 | 10.643 | 11.558 | 12.386 | 13.275 | 14.014 |
| *G_inco (Ours)* | 0.251 | 0.3462 | 0.4315 | 0.4976 | 0.5529 | 0.6112 | 0.6480 | 0.6983 | 0.7424 |

Table 2: $m_{tr} = 15, m_{va} = 1$

| epoch | 20 | 40 | 60 | 80 | 100 | 120 | 140 | 160 | 180 |
|---|---|---|---|---|---|---|---|---|---|
| *Train-Test gap* | 0.06857 | 0.03689 | 0.09728 | 0.0327 | 0.1113 | 0.1628 | 0.04986 | 0.06426 | 0.1674 |
| *Lipschitz* | 55.79 | 77.96 | 95.09 | 109.6 | 122.4 | 133.9 | 144.6 | 154.5 | 163.8 |
| *G_norm* | 17.29 | 21.39 | 25.07 | 28.04 | 30.24 | 32.79 | 35.15 | 37.63 | 39.73 |
| *G_inco (Ours)* | 0.1774 | 0.2486 | 0.3127 | 0.3661 | 0.4074 | 0.4481 | 0.4817 | 0.5182 | 0.5468 |

Table 3: $m_{tr} = 1, m_{va} = 15$

| epoch | 20 | 40 | 60 | 80 | 100 | 120 | 140 | 160 | 180 |
|---|---|---|---|---|---|---|---|---|---|
| *Train-Test gap* | 0.2389 | 0.06293 | 0.3155 | 0.1718 | 0.1822 | 0.2196 | 0.1687 | 0.1814 | 0.1923 |
| *Lipschitz* | 14.21 | 19.86 | 24.22 | 27.91 | 31.17 | 34.12 | 36.83 | 39.35 | 41.72 |
| *G_norm* | 4.484 | 5.579 | 6.572 | 7.363 | 7.953 | 8.632 | 9.249 | 9.899 | 10.42 |
| *G_inco (Ours)* | 0.7801 | 1.048 | 1.286 | 1.45 | 1.617 | 1.76 | 1.89 | 2.018 | 2.149 |

Where Train-Test gap is the observed generalization error, G_inco is the whole gradient incoherence bound, i.e: $\sqrt{\frac{\sigma^2(\epsilon_U + \epsilon_W)}{nm_{va}}}$, G_norm is the corresponding bound w.r.t. gradient norm.

Thus, we can see that the gradient-incoherence bound is much closer to the estimation of the actual gap but can be improved in the future.

### D.2 Omniglot

Now we give additional experimental results for the deep few-shot benchmark – Omniglot. We compare the test accuracy for Meta-SGLD with three train-validation split settings, *i.e.*, $m_{va} = \{1, 8, 15\}$. The test accuracy for MAML and Meta-SGLD with $\{0, 1, 4, 10\}$ fine-tune steps are illustrated in Table 4.

Under the same experiment settings, Meta-SGLD achieves slightly better performance than our reproduced MAML. However, our test accuracy is not comparable to the original results of MAML [2].

We only trained the model with 2000 epochs, and the other hyper-parameter settings are also different from [2]. Moreover, our Meta-SGLD code is modified based on [48]. This realization version of MAML is claimed by the author to have worse performance than original MAML. We would like to re-emphasize that our experiments were conducted to validate our theories but not to achieve SOTA results.

Comparing experimental results for different train-validation split settings, we note that the train loss at last epoch for $m_{va} = 1$ is smaller than $m_{va} = 8$, while the best test accuracy is obtained with $m_{va} = 8$. Non-rigorously we think the generalization error of $m_{va} = 8$ should be smaller than $m_{va} = 1$. The consistent result was verified by the gradient-incoherence bound, which is the tightest for $m_{va} = 8$. For $m_{va} = 15$, *i.e.*, training with 1-shot data, both the test accuracy, train loss and the estimated bound were the worst.

Table 4: Test Accuracy for Omniglot, train with 2000 epochs

| 5-way Test Accuracy | | | |
|---|---|---|---|
| Algorithm | $m_{va} = 15$ | $m_{va} = 8$ | $m_{va} = 1$ |
| MAML 0-step | 20.7% | 20.13% | 20.26% |
| MAML 1-step | 88.43% | 95.8% | 92.43% |
| MAML 4-step | 90.77% | 96.97% | 96.14% |
| MAML 10-step | 91.06% | 97.07% | 96.53% |
| Meta-SGLD 0-step | 19.48% | 20.06% | 20.29% |
| Meta-SGLD 1-step | 88.8% | 95.95% | 92.8% |
| Meta-SGLD 4-step | 91.1% | 96.97% | 96.1% |
| Meta-SGLD 10-step | 91.26% | 97.1% | 96.53% |

**Comparison with the observed generalization error**

Similar to the synthetic setting, we calculated the observed generalization error by evaluating the expected difference between the train loss and test loss. And we list the results of Omniglot data under different train-validation split setting in the following Table 5, 6 and 7:

Table 5: $m_{tr} = 8, m_{va} = 8$

| epoch | 200 | 400 | 600 | 800 | 1000 | 1200 | 1400 | 1600 | 1800 |
|---|---|---|---|---|---|---|---|---|---|
| *Train-Test gap* | 0.01896 | 0.00364 | 0.008821 | 0.01856 | 0.01366 | 0.0001578 | 0.04087 | 0.02269 | 0.01669 |
| *Lipschitz* | 4.8159 | 6.8108 | 8.3415 | 9.6319 | 10.7688 | 11.7966 | 12.7418 | 13.6215 | 14.4478 |
| *G_norm* | 0.1835 | 0.2765 | 0.3578 | 0.4292 | 0.4959 | 0.5557 | 0.6134 | 0.6679 | 0.7204 |
| *G_inco (Ours)* | 0.109 | 0.1372 | 0.1617 | 0.1841 | 0.2057 | 0.2252 | 0.2444 | 0.2625 | 0.2798 |

Table 6: $m_{tr} = 15, m_{va} = 1$

| epoch | 200 | 400 | 600 | 800 | 1000 | 1200 | 1400 | 1600 | 1800 |
|---|---|---|---|---|---|---|---|---|---|
| *Train-Test gap* | 0.01508 | 0.02493 | 0.1099 | 0.07129 | 0.01425 | 0.07677 | 0.007417 | 0.04808 | 0.04199 |
| *Lipschitz* | 9.3429 | 13.2129 | 16.1824 | 18.6858 | 20.8914 | 22.8854 | 24.719 | 26.4258 | 28.0287 |
| *G_norm* | 0.4536 | 0.6265 | 0.8012 | 0.9641 | 1.115 | 1.265 | 1.409 | 1.55 | 1.688 |
| *G_inco (Ours)* | 0.123 | 0.1572 | 0.2146 | 0.2787 | 0.3437 | 0.4154 | 0.4775 | 0.5275 | 0.5848 |

Table 7: $m_{tr} = 1, m_{va} = 15$

| epoch | 200 | 400 | 600 | 800 | 1000 | 1200 | 1400 | 1600 | 1800 |
|---|---|---|---|---|---|---|---|---|---|
| *Train-Test gap* | 0.0474 | 0.001597 | 0.02478 | 0.01946 | 0.02008 | 0.006832 | 0.05856 | 0.09882 | 0.0331 |
| *Lipschitz* | 45.8866 | 64.8935 | 79.4779 | 91.7732 | 102.6056 | 112.3988 | 121.4046 | 129.7869 | 137.6598 |
| *G_norm* | 0.9537 | 0.9639 | 0.9756 | 0.9861 | 0.9974 | 1.011 | 1.025 | 1.039 | 1.053 |
| *G_inco (Ours)* | 0.9534 | 0.9619 | 0.971 | 0.9785 | 0.9863 | 0.9959 | 1.006 | 1.015 | 1.025 |

Where Train-Test gap is the observed generalization error, G_inco is the whole gradient incoherence bound, i.e: $\sqrt{\frac{\sigma^2(\epsilon_U + \epsilon_W)}{nm_{va}}}$, G_norm is the corresponding bound w.r.t. gradient norm.

# E   Experiment Details

Although we have described the detailed algorithm in the main paper to obtain a data-dependent estimate bound, we offer a more structural pseudo-code in section G. We used Monte Carlo simulations to estimate our generalization error bound in Theorem 6.2. Recall the accumulated gradient incoherence for meta learner and base learner are respectively denoted as:

$$\epsilon_U = \sum_{t=1}^{T} \mathbb{E}_{B_{I_t}^{\text{va}}, B_{I_t}^{\text{tr}}, W_{I_t}, U^{t-1}} \frac{\eta_t \gamma_t \|\epsilon_t^u\|_2^2}{2}, \quad \epsilon_W = \sum_{t=1}^{T} \sum_{i=1}^{|I_t|} \sum_{k=1}^{K} \mathbb{E}_{B_{i,t,k}^{\text{va}}, B_{i,t,k}^{\text{tr}}, W_{i,t}^{k-1}} \frac{\beta_{t,k} \gamma_{t,k} \|\epsilon_{t,i,k}^w\|_2^2}{2}.$$

In our experiments, the two terms are separately estimated. Since we have

$$\tilde{R}_{B_{I_t}^{\text{va}}}(U^{t-1}) = \frac{1}{|I_t|} \sum_{i \in I_t} R_{B_{i,t}^{\text{va}}}(W_{i,t}^K),$$

$\epsilon_t^u = \nabla \tilde{R}_{B_{I_t}^{\text{va}}, B_{I_t}^{\text{tr}}}(U^{t-1}) - \nabla \tilde{R}_{B_{I_t}^{\text{tr}}}(U^{t-1})$ is related to the last inner step output $W_{i,t}^K$. To estimate $\epsilon_U$, we conducted 10 times Monte Carlo simulations for the corresponding inner path at each iteration $t$, the gradients are calculated with back-propagation. For $\epsilon_W$, it's much simpler, we just conducted 10 times Monte Carlo simulations at each inner step, see more details in the code. Our code is modified based on Long [48] and Amit [49].

## E.1   Synthetic Data

**Network Structure** For Synthetic Data, the model structure is quite simple. It is a 2D mean estimation. For a single task with parameter $w$, to estimate the mean, we need to calculate the loss $\ell(W, Z) = \|W - Z\|_2^2$. Hence, we constructed a single layer that conducts $W - Z$. Then the output of this layer and the pseudo target (always set to 0) were taken as input to a square loss function.

| Hyper parameters | values |
|---|---|
| task numbers $n$ | 20000 |
| sample numbers $m$ | 16 |
| outer Loop inverse temperature $\gamma_t$ | 10000 |
| Inner Loop inverse temperature $\gamma_{t,k}$ | 10000 |
| Outer Loop learning rate $\eta_t$ | 0.2 |
| Inner Loop learning rate $\beta_{t,k}$ | 0.4 |
| task batch size $|I_t|$ | 5 |
| epoch/Outer Loop iterations $T$ | 200 |
| Inner Loop updates $K$ | 4 |
| $m_{\text{va}}$ | $\{1, 8, 15\}$ |
| $m_{\text{tr}}$ | $\{15, 8, 1\}$ |
| data dimension | 2 |
| loss | square loss |
| test update step | 10 |

Table 8:  Synthetic Data Experiment Setting

**Training Details** The hyper parameter settings and training details for Synthetic data set are presented in Table 8.

**Compute Resource** All experiments for Synthetic data were tested on a machine runing macOS system with an Intel Core i5 CPU, 8G memory.

**Subgaussian parameter** For the synthetic data, we want to estimate the mean for each sub-task, where we have for task $i$, $Z \sim \mathcal{N}(\mu_i, 0.1\mathbb{I}_d), d = 2$. The task mean $\mu_i$ is sampled from the truncated normal distribution $\mathcal{N}((-4, -4)^T, 5\mathbb{I}_2)$ with $\mu_i \in [-12, 4] \times [-12, 4]$. Thus we have $\|\mu_i\|_2^2 \leq 288$. $\tilde{W}$ is the independent copy of the SGLD algorithm output $W$. To estimated the $\sigma^2$ that satisfies the subgaussian loss, We consider the worst case where the output is obtained with a single example $Z'$ and one inner step update. So we have $W = \mathbf{0} - 2\beta(\mathbf{0} - Z') + \epsilon \approx 0.8Z'$, since the inner loop learning rate in our experiment setting is $0.4$(the noise added is quite small, which can be ignored). Hence, $\tilde{W} \sim \mathcal{N}(0.8\mu_i, 0.064\mathbb{I}_d)$. Moreover, we have $\tilde{W} \perp\!\!\!\perp Z$ and $\ell(\tilde{W}, Z) = \|\tilde{W} - Z\|_2^2$, so

$\tilde{W} - Z \sim \mathcal{N}(0.2\mu_i, \sigma_l^2 \mathbb{I}_d), \sigma_l^2 = 0.164$. Furthermore, $\ell(\tilde{W}, Z) \sim \sigma_l^2 \prime\chi_d^2(k), k = 0.04||\mu_i||_2^2$, which is a noncentral chi-squared distribution(Bu et al. [37] analyzed ERM, where $\ell(\tilde{W}, Z)$ follows central chi-squared distribution). Thus the CGF of $\ell(\tilde{W}, Z)$ is given by:

$$\psi_{\ell(\tilde{W},Z)}(\lambda) = -(d+k)\sigma_l^2\lambda - \frac{d}{2}\log(1 - 2\sigma_l^2\lambda) + \frac{k\sigma_l^2\lambda}{1 - 2\sigma_l^2\lambda}$$

$$= \frac{d}{2}(-2\sigma_l^2\lambda - \log(1 - 2\sigma_l^2\lambda)) + k\sigma_l^2\lambda\frac{2\sigma_l^2\lambda}{1 - 2\sigma_l^2\lambda}, \lambda \in (-\infty, \frac{1}{2\sigma_l^2})$$

Let $u \overset{\text{def}}{=} 2\sigma_l^2\lambda$, and note that $-u - \log(1 - u) \le \frac{u^2}{2}, u < 0$.

$$\psi_{\ell(\tilde{W},Z)}(\lambda) = \frac{d}{2}(-u - \log(1 - u)) + \frac{ku^2}{2(1 - u)} \le \frac{du^2}{4} + \frac{ku^2}{2} = (2k + d)\sigma_l^4\lambda^2, \lambda < 0.$$

So the subgaussian parameter $\sigma^2$ in our assumption can be expressed as $\sigma^2 = 2(2k + d)\sigma_l^4 = 2(2 * 0.04||\mu_i||_2^2 + d)(0.164)^2$, where $d = 2$ and $||\mu_i|| \le 288$. So we obtain $\sigma^2 = 0.164 * 0.164 * 4 * (1 + 0.04 * 288) = 1.3469$.

### E.2 Omniglot

**Network Structure** We used a CNN network architecture for Omniglot data set, which consists of a stack of modules. The first three modules are the same, each of which is a $3 \times 3$ 2d convolution layer of 64 filters and stride 2 followed by a Relu layer and a batch normalization layer. Then the fourth module is a $2 \times 2$ 2d convolution layer of 64 filters and stride 1, followed by a Relu layer and a batch normalization layer. Through the aforementioned modules, we got a $64 \times 1 \times 1$ feature map. This feature map was further taken into a fully connected layer which output the logits for a 5-way classification. Finally, the cross-entropy loss is calculated with the logits and the corresponding labels.

**Training Details** The hyper parameter settings and training details for Omniglot data set are outlined in Table 9.

**Compute Resource** The experiments for Omniglot were run on a server node with 6 CPUs and 1 GPU of 32GB memory.

**Subgaussian parameter** For Omniglot data, we used the cross entropy loss, which is unbounded. And the data distribution is too complex that we cannot obtain a similar closed form estimation for the subgaussian parameter. To assure the theoretic guarantee, we can adopt a variation of the loss function which is clipped to $[0, 2]$ and hence 1-subgaussian. Actually, such clip is not always necessary. As we discussed in section C, the subgaussian parameter $\sigma^2$ is related to the independent copy $\tilde{W}$ of the base learner output for each task. During our experiments, the loss w.r.t $\tilde{W}$ rarely exceed the clip value.

## F Additional Comparison to Related Works

**Discussion with Jose and Simeone [20]** They adopted different and generally unrealistic assumptions to derive the theoretical results. Concretely:

In joint-training (Eq (33) in Jose and Simeone [20]), the task-level error w.r.t. base-learner $W$ is related to the unknown environment distribution $P_T$, which is hard to estimate from the observed data. In contrast, the task-level risk in our paper is associated with the distribution meta-parameter $U$, which can be evaluated efficiently. Besides, when $m \to \infty$ and the number of task $n$ is limited, their bound always has a non-zero term. This does not fit the reality since the new task already has enough samples to learn.

In the alternate-training (meta train-validation) settings, they assumed the task parameters $W$ and $S^{va}$ are conditionally independent given $S^{tr}$ (Eq A(8) in their paper). This is an unrealistic condition in meta-learning since $W$ depends on the meta-parameter $U$, where $U$ is updated by $S_{1:n}^{va}$. As a result, if we set $m = 1$ (each task has only one sample), then $n \to \infty$, the upper bound in Eq(3) of [20] will converge to 0, which is problematic since task distribution can be arbitrary noisy and the task-level

| Hyper parameters | values |
|---|---|
| task numbers $n$ | $\binom{1200}{5}$ |
| sample numbers $m$ | 16 |
| outer Loop inverse temperature $\gamma_t$ | 100000000 |
| Inner Loop inverse temperature $\gamma_{t,k}$ | 100000000 |
| Outer Loop learning rate $\eta_t$ | $10^{-3} * 0.96^{\frac{t}{800}}$ |
| Inner Loop learning rate $\beta_{t,k}$ | $0.3 * 0.96^{\frac{t}{1000}}$ |
| n-way classification | 5 |
| task batch size $|I_t|$ | 32 |
| epoch/Outer Loop iterations $T$ | 2000 |
| Inner Loop updates $K$ | 4 |
| $m_{\text{va}}$ | $\{1, 8, 15\}$ |
| $m_{\text{tr}}$ | $\{15, 8, 1\}$ |
| loss | cross entropy |
| test update step | 10 |
| image size | 28*28 |
| image channel | 1 |

Table 9: Omniglot Experiment Setting

error (with one sample) can be quite large. Besides, this bound is irrelevant to the train validation split, which is inconsistent with the previous work such as [33, 35].

Therefore, our theoretical results are not directly comparable. Even if we ignore all these unrealistic theoretical assumptions and directly compare the results in Jose and Simeone [20], their theoretical results in noisy iterative approaches still depend on the Lipschitz constant of the neural network (Eq (45) in their paper), which is vacuous in deep learning.

**Discussion with recent theoretical analysis on the support-query approach**

Denevi et al. [33] first studied train-validation split for meta-learning in biased linear regression model. They proved a generalization bound and concluded that there exists a trade-off for train-validation split, which is consistent with Theorem 5.2 in our paper. Specifically, they constructed two datasets: For the simple unimodal distribution, the optimal split is $m_{tr} = 0$. For the bimodal distribution, the optimal split is $m_{tr} \in (0, m-1]$.

Bai et al. [34] proposed a theoretical analysis of train-validation split in linear centroid meta-learning (parameter transfer). By comparing the train-val (alternate training) and train-train (joint training) method, they showed that train-validation split is necessary for the agnostic setting, where the train-val meta loss is an unbiased estimator w.r.t. the meta-test loss while the train-train loss is biased(consistent with our Theorem 5.1). When it is realizable (noiseless scenario), the train-train model can achieve better excess loss.

Saunshi et al. [35] analyze the train-valid splitting for linear representation learning (representation transfer). They proved that the train-validation split encourages learning a low-rank representation. In the noiseless setting, the train-val method already enables low-rank representation, so it's preferable to set a smaller train-split and larger validation-split.

While our work focus on general settings with randomized algorithms and does not specify the form of base-learner and meta-learner, which can be applied in non-linear representation, non-linear classifier, and non-convex loss. Besides, the relations of our papers are as follows:

1. Our theory can recover the stochastic version of the above parameter and representation transfer settings. If we consider the linear model with $\mathcal{U} = \mathcal{W} \subseteq R^d$ and $P_{W|U}$ is approximated by a Gaussian distribution $\mathcal{N}(U, \mathbb{I}_d)$, the problem is analogous to parameter-transfer meta-learning. If $\mathcal{U} \subseteq R^k, \mathcal{W} \subseteq R^{k+d}$ (where $U \in R^k$ is the shared representation parameter, $V \in R^d$ is the parameter of the linear classifier, $W = (U, V) \in R^{(k+d)}$ is the whole task parameter) and the prior of stochastic linear classifier $V$ is approximated by a Gaussian distribution $\mathcal{N}(0, \mathbb{I}_d)$, the setting is similar to the representation transfer paradigm.

2. Since our bounds are based on the generic settings (flexible data distribution, algorithm, and loss choice), the two training modes are not directly comparable in our problem. However, we agree on

the potential limit of joint training (asymptotically biased in the agnostic setting) and believe it is highly interesting to explore the specific conditions to understand the benefits and limitations of these training modes as the future work.

# G   Pseudo Code

---

**Algorithm 1:** Meta-SGLD for Few-Shot Learning

---

**Require**: Task environment $\tau$;
**Require**: initial learning rates $\eta_0, \beta_0$, inverse temperature $\gamma$;
randomly initialize $U^0$;
**for** $t \leftarrow 1$ *to* $T$ **do**
    Sample task data batch $B_i \sim \mu_{m,\tau}, \forall i \in I_t$;
    Randomly split $B_{I_t}$ to $B_{I_t}^{tr}$ and $B_{I_t}^{va}$;
    learning rate decay, get $\eta_t$;
    **for** $i \leftarrow 1$ *to* $|I_t|$ **do**
        **for** $k \leftarrow 1$ *to* $K$ **do**
            learning rate decay, get $\beta_{t,k}$;
            **if** *GLD* **then**
                Use full batch, $B_{i,t,k}^{tr} = B_{i,t}^{tr}$ and $B_{i,t,k}^{tr} = B_{i,t}^{va}$;
            **else**
                Sample $B_{i,t,k}^{tr}$ from $B_{i,t}^{tr}$;
                Sample $B_{i,t,k}^{va}$ from $B_{i,t}^{va}$;
            **end**
            Update parameter with gradient descent:;
            **if** $k == 1$ **then**
                $W_{i,t}^{k-1} = U^{t-1}$;
            **end**
            Calculate $\mathbb{E}_{B_{i,t,k}^{\text{va}}, B_{i,t,k}^{\text{tr}}, W_{i,t}^{k-1}} \frac{\beta_t^k \gamma_t^k \|\epsilon_{t,i,k}^w\|_2^2}{2}$ with Monte Carlo simulation;
            $W_{i,t}^k = W_{i,t}^{k-1} - \beta_{t,k} \nabla R_{B_{i,t,k}^{\text{tr}}}(W_{i,t}^{k-1}) + \zeta^{t,k}$;
        **end**
    **end**
    Calculate $\mathbb{E}_{B_{I_t}^{\text{va}}, B_{I_t}^{\text{tr}}, U^{t-1}, W_{I_t}} \frac{\eta_t \gamma_t \|\epsilon_t^u\|_2^2}{2}$ with Monte Carlo Simulation;
    $U^t = U^{t-1} - \eta_t \nabla \frac{1}{|I_t|} \sum_{i \in I_t} R_{B_{i,t}^{\text{va}}}(W_{i,t}^K) + \xi^t$
**end**

---