# OpenReview forum: "Generalization Bounds For Meta-Learning: An Information-Theoretic Analysis"
_NeurIPS.cc/2021/Conference — NeurIPS 2021 Spotlight_

### Official Review · Reviewer_v93c · 2021-07-05

**Rating:** 7
**Confidence:** 4

**Summary:**

The authors derive two bounds on the generalization property of meta-learning algorithms using an information-theoretic analysis. The first one considers algorithms based on a *joint-training*, i.e.  when all the data from the tasks is used to update the parameters. The second bound is specific to algorithms learned through an *alternate training*, i.e. the tasks data are split into training and validation, training part is used to update the task-specific parameters and the validation part is used to update the meta-parameters.
Then, they analyze two concrete algorithms based on SGLD, a joint-training version and an alternate training version, to derive the bounds for these specific cases.
Finally, they empirically estimate the values of their bounds and compare with an empirical estimate of a previous bound based on the lipschitz constant of the network. The empirical estimation is done on synthetic data and the omniglot dataset.

**Limitations And Societal Impact:**

The work presented here is very theoretical. To study the possible negative impacts of this kind of work is to study the negative impacts of the whole field. On the other hand, contrary to what the authors say, meta-learning could also be used for harmful applications such as quick racial profiling of a user.

**Main Review:**

__**Originality:**__  The authors specifically studied the generalization of meta-learning algorithms through the lens of information theory. They obtained bounds depending on the mutual information between the data and the outputs of the meta-learner and the base learner, similar to what was obtained in the general case of any randomized algorithm (Xu and Raginsky [15]). The authors discuss their results and compare them with other previously obtained bounds. However, I think that the authors could emphasize more the differences in their approach that allowed them to obtain different results. The only explanation of the specificity of their approach is detailed in the proof of theorem 5.1 in the supplementary material. If I understand correctly, they do not separate the environment error and the task error, which allows them to have a tighter bound.

__**Quality:**__  The approach used by the authors seems correct, and the bounds obtained are tighter than previous results. The authors provide in the experimental section an *estimation of the values* obtained in their bounds in practice and compare them with an *estimate of the values* used in previous results. The authors have thus shown that, empirically, the values of the quantities involved in their generalization bounds are smaller than those of previous results that use the Lipschitz norm of the model. This is not an empirical validation of the theory, which would show that the bounds fit with the experiments.

__**Clarity:**__  The paper is very well written and well-structured, easy to read and follow. The results obtained are well detailed and compared w.r.t previous work. The explanations of the differences between the analysis done by the authors and previous analyses could be improved (as explained above).

__**Significance:**__ The bounds obtained seem empirically tighter than previous bounds. It remains to be shown that they actually fit with the experiment. As mentioned by the authors, the analysis carried out here could inspire new methods, but it is not the object of the paper.

## Minor Remarks

- The meaning of the independent symbol between two sets should be explicitly defined (l.146).
- I think Fig.1 should be moved higher in the paper. It is mentioned on page 4 and doesn't appear until page 6. One must scroll down to see the figure the first time it is mentioned.
- Some minor typos I noticed:
  - l.59 "different ~~the~~ *from* those"
  - l.81 "~~the~~ the"
  - l.94 "~~of~~ of"
  - l.342 "*~~h~~as* compared to"

**Time Spent Reviewing:**

I spent between 8 to 10 hours to review this paper.

---

> ### Author Response · Authors · 2021-08-10
> **Response**
>
> We thank the reviewer for appreciating the contribution of our work, and we are grateful for the suggestions, which indeed help improve our work. Below are our responses to the main comments:
>
> ###### **“emphasize more the differences in their approach that allowed them to obtain different results….The explanations of the differences between the analysis done by the authors and previous analyses”**
>
>
> We agree with your comments and will add related discussions in our final version.
>
> - **Comparison with single task mutual information-theoretical results**
>
>    * The proofs in Theorem 5.1 and 5.2 are based on the decoupling estimate lemma, which has been used in many single-task mutual information-theoretic learning works. However, extending the proof idea to the meta-learning is quite challenging since it involves understanding the risk bound w.r.t both meta- and base-parameters.
>    * Moreover, our theoretical results can induce a meaningful (or non-vacuous) generalization bound w.r.t. MAML algorithm, which has been widely adopted in deep learning. Several previous works have derived the theoretical analysis, while their results depend on some constants that can be pretty large in neural networks. e.g., in the gradient-based few-shot learning bound, most theoretical results rely on the Lipschitz constant (the maximum of the gradient norm) w.r.t. the DNN, which can be $1.7 \times 10^6$ in the Omniglot. Therefore, the previous bound did not provide a meaningful interpretation of the generalization property in MAML. For a more meaningful comparison, we used the expected gradient norm bound as a tighter estimation of Lipchitz bound in our paper. As for the additional theoretical work related to gradient norm based theory, we refer to Mou et al.[2018], which focused on single-task learning.
>
> ###### **“ ...they do not separate the environment error and the task error, which allows them to have a tighter bound.”**
>
> Yes. The key technique herein is to consider the meta and task parameters as a joint term and exploit their probabilistic relation to obtain our results.
>
>
> ###### **“show that the bounds fit with the experiments.”**
>
> If we understand correctly, the reviewer wants us to show the inequality of the Theorem is correct. Hence, we calculated the actual generalization gap by evaluating the expected difference between the train loss and test loss. We list the results of synthetic data in the following tables:
>
> - $m_{tr}=8, m_{va}=8$:
>
> | epoch | 20 | 40 | 60 | 80 | 100 | 120 | 140 | 160 | 180 |
> | --- | --- | --- | --- | --- | --- | --- | --- | --- | --- |
> | Train-Test gap | 0.0697 | 0.0371 | 0.0908 | 0.0241 | 0.1072 | 0.1492 | 0.1775 | 0.1432 | 0.1581 |
> | G_norm | 6.009 | 7.506 | 8.807 | 9.856 | 10.643 | 11.558 | 12.386 | 13.275 | 14.014 |
> | G_inco (Ours) | 0.251 | 0.3462 | 0.4315 | 0.4976 | 0.5529 | 0.6112 | 0.6480 | 0.6983 | 0.7424 |
> |Lipschitz |19.77|27.63|33.7|38.84|43.37|47.47|51.24|54.75|58.05|
>
> - $m_{tr}=15, m_{va}=1$:
>
> | epoch | 20 | 40 | 60 | 80 | 100 | 120 | 140 | 160 | 180 |
> | --- | --- | --- | --- | --- | --- | --- | --- | --- | --- |
> | Train-Test gap |0.06857|0.03689|0.09728|0.0327|0.1113|0.1628|0.04986|0.06426|0.1674|
> | G_norm | 17.29|21.39|25.07|28.04|30.24|32.79|35.15|37.63|39.73|
> | G_inco (Ours) | 0.1774|0.2486|0.3127|0.3661|0.4074|0.4481|0.4817|0.5182|0.5468|
> | Lipschitz |55.79|77.96|95.09|109.6|122.4|133.9|144.6|154.5|163.8|
>
> - $m_{tr}=1, m_{va}=15$:
>
> | epoch | 20 | 40 | 60 | 80 | 100 | 120 | 140 | 160 | 180 |
> | --- | --- | --- | --- | --- | --- | --- | --- | --- | --- |
> | Train-Test gap |0.2389|0.06293|0.3155|0.1718|0.1822|0.2196|0.1687|0.1814|0.1923|
> | G_norm | 4.484|5.579|6.572|7.363|7.953|8.632|9.249|9.899|10.42|
> | G_inco (Ours)| 0.7801|1.048|1.286|1.45|1.617|1.76|1.89|2.018|2.149|
> | Lipschitz |14.21|19.86|24.22|27.91|31.17|34.12|36.83|39.35|41.72|
>
>
> Where Train-Test gap is the actual generalization gap, G_inco is the whole gradient incoherence bound, i.e: $\sqrt{\frac{\sigma^2(\epsilon_U + \epsilon_W)}{nm_{va}}}$, G_norm is the corresponding bound w.r.t. gradient norm.
>
> We also have added the estimation of Lipchitz bound, and the result validates that it's rather vacuous. The actual generalization gap is smaller than the gradient incoherence bound, which validates our theory. The above experiment result further suggests that the gradient-incoherence bound is indeed much better but can be improved in the future.
>
> Thanks again for your suggestions to improve the paper. We will fix mentioned typos and adjust the format in the final version.
>
> #### **References**
>
> Mou, Wenlong, et al. "Generalization bounds of sgld for non-convex learning: Two theoretical viewpoints." Conference on Learning Theory. PMLR, 2018.

---

> > ### Comment · Reviewer_v93c · 2021-08-27
> > **Thanks for the additional experiments**
> >
> > I thank the authors for their detailed response and the additional experiments. In light of these experiments which addressed my main concern, I'm raising my score.

---

> > > ### Author Response · Authors · 2021-08-29
> > > **Thanks for your feedback**
> > >
> > > We appreciate the reviewer's suggestions and we will update our final version according to your feedback.

---

### Official Review · Reviewer_zirw · 2021-07-08

**Rating:** 8
**Confidence:** 2

**Summary:**

The paper extends information-theoretic bounds (Xu and Raginsky [15]) to provide generalization bounds for two meta-learning settings: Joint Training (classical setting of Baxter) and Alternate Training (that includes the MAML algorithm). The general bounds are based on the mutual information between the learning algorithm output and input. The bound is given a more explicit form for the case of a meta-SGLD algorithm, that can be described as a stochastic variant of the popular MAML algorithm. Experimental results show bounds that are orders of magnitudes smaller than norm-based bounds.

**Limitations And Societal Impact:**

The limitations are discussed in the main review. There is no potential negative societal impact.

**Main Review:**

 Strengths:

- The results of Theorem 5.1, 5.2, and 6.2  seem very interesting, as they provide a new viewpoint to analyze the generalization properties of meta-learning algorithms.
In particular, Thm 5.2 sheds light on meta-learning settings with dataset split and shows a trade-off in choosing the split sizes.
- The Meta-SGLD algorithm and its generalization guarantees seem significant since it depends on the gradient differences rather than on the Lipschitz constant of the loss (as in previous work), which can be very large in deep learning settings.
****************************
Weaknesses:

- The main paper is well-written and easy to follow. However, the proofs are harder to follow (especially for Thm 5.1 and 5.2).
I suggest starting the proofs with a high-level explanation and giving motivation to each step.

- The bounds in the experiments are not compared to the actual generalization gap. The figures show a generalization gap as small as 0.1. I think it is important to validate that the actual generalization gap is smaller than that.

- The clarity of the experiment should be improved.  The figures lack axes annotations. I couldn’t see a description of what is plotted. What exactly is the G_Norm bound? How is it evaluated?

Comments:
- Thm 6.1 and 6.2 show a trade-off in choosing $T$. It would be interesting to discuss and investigate it more.

****************************

Minor comments:
- Thm 5.2: $I^{S^{tr}}$ is undefined (only defined in the appendix)




**Time Spent Reviewing:**

9

---

> ### Author Response · Authors · 2021-08-10
> **Response**
>
> We thank the reviewer for appreciating our theoretical insights and the strength of our work. In addition, we appreciate the reviewer’s effort to provide constructive comments which we can incorporate in the revision. Below are our responses to your comments.
>
> #### **High-level explanation of the proof**
>
> Thanks for your suggestions. Additional descriptions of the high-level proof idea and motivations for each step will be updated in the final version.
> In the high-level idea, Theorem 5.1 and 5.2 are based on the decoupling estimate lemma, which has been used in information-theoretic single-task learning. Different from previous conventional proof logic in PAC Bayes meta-learning(which is hard to extend to alternate training) that adopted two-level risk bound and then combined two-level bounds through union bound (e.g., Pentina et al.[2014] and Amit et al.[2018]), we consider a novel proof idea by treating the meta parameter and task parameters as a joint term. Exploiting their probabilistic relation enables us to obtain interpretable and practically meaningful results.
>
> #### **Experiment explanation**
>
> G_Norm bound is indeed a tighter approximation of Lipschitz bound. Since the Lipschitz constant is defined as the most significant gradient norm w.r.t. the neural network, this term can be extremely large in deep neural networks. We also empirically observed the largest gradient norm (Lipschitz constant) as $1.7 \times 10^6$ in the Omniglot. Therefore we compare its tighter version in the paper: the expected gradient norm w.r.t. the neural-network parameters. As for the additional theoretical work related to G_norm-based generalization error, we refer to Mou et al.[2018], which focused on single-task learning.
>
>
> #### **Actual generalization gap**
>
> We calculated the actual generalization gap by evaluating the expected difference between the train loss and test loss.  And we list the results of synthetic data in the following tables:
>
>
> - $m_{tr}=8, m_{va}=8$:
>
> | epoch | 20 | 40 | 60 | 80 | 100 | 120 | 140 | 160 | 180 |
> | --- | --- | --- | --- | --- | --- | --- | --- | --- | --- |
> | Train-Test gap | 0.0697 | 0.0371 | 0.0908 | 0.0241 | 0.1072 | 0.1492 | 0.1775 | 0.1432 | 0.1581 |
> | G_norm | 6.009 | 7.506 | 8.807 | 9.856 | 10.643 | 11.558 | 12.386 | 13.275 | 14.014 |
> | G_inco (Ours) | 0.251 | 0.3462 | 0.4315 | 0.4976 | 0.5529 | 0.6112 | 0.6480 | 0.6983 | 0.7424 |
> |Lipschitz |19.77|27.63|33.7|38.84|43.37|47.47|51.24|54.75|58.05|
>
> - $m_{tr}=15, m_{va}=1$:
>
> | epoch | 20 | 40 | 60 | 80 | 100 | 120 | 140 | 160 | 180 |
> | --- | --- | --- | --- | --- | --- | --- | --- | --- | --- |
> | Train-Test gap |0.06857|0.03689|0.09728|0.0327|0.1113|0.1628|0.04986|0.06426|0.1674|
> | G_norm | 17.29|21.39|25.07|28.04|30.24|32.79|35.15|37.63|39.73|
> | G_inco (Ours) | 0.1774|0.2486|0.3127|0.3661|0.4074|0.4481|0.4817|0.5182|0.5468|
> | Lipschitz |55.79|77.96|95.09|109.6|122.4|133.9|144.6|154.5|163.8|
>
> - $m_{tr}=1, m_{va}=15$:
>
> | epoch | 20 | 40 | 60 | 80 | 100 | 120 | 140 | 160 | 180 |
> | --- | --- | --- | --- | --- | --- | --- | --- | --- | --- |
> | Train-Test gap |0.2389|0.06293|0.3155|0.1718|0.1822|0.2196|0.1687|0.1814|0.1923|
> | G_norm | 4.484|5.579|6.572|7.363|7.953|8.632|9.249|9.899|10.42|
> | G_inco (Ours)| 0.7801|1.048|1.286|1.45|1.617|1.76|1.89|2.018|2.149|
> | Lipschitz |14.21|19.86|24.22|27.91|31.17|34.12|36.83|39.35|41.72|
>
>
> Where Train-Test gap is the estimation of the actual generalization gap, G_inco is the whole gradient incoherence bound, i.e: $\sqrt{\frac{\sigma^2(\epsilon_U + \epsilon_W)}{nm_{va}}}$, G_norm is the corresponding bound w.r.t. gradient norm.
>
> We also have added the estimation of Lipchitz bound, and the result validates that it's vacuous. The actual generalization gap is smaller than the gradient incoherence bound. The above experiment result further suggests that the gradient-incoherence bound is indeed much better but can be improved in the future.
>
> #### **Other points**
> We will fix the figure format and provide an additional description in the final version. Moreover, we agree the choice of $T$ is also an interesting direction.
>
> ### **References**
> Ankit Pensia, Varun Jog, and Po-Ling Loh. Generalization error bounds for noisy, iterative algorithms. In 2018 IEEE International Symposium on Information Theory (ISIT), pages 546–550. IEEE, 2018.
>
> Mou, Wenlong, et al. "Generalization bounds of sgld for non-convex learning: Two theoretical viewpoints." Conference on Learning Theory. PMLR, 2018.

---

> > ### Comment · Reviewer_zirw · 2021-08-17
> > **Response to authors**
> >
> > I thank the authors for their detailed response.  The authors had adequately answered my points. I raised the score accordingly.

---

> > > ### Author Response · Authors · 2021-08-17
> > > **Thanks for your feedback**
> > >
> > > We appreciate the reviewer's suggestions that help us make the work more complete. And we will update our final version according to your feedback.

---

### Official Review · Reviewer_YsVe · 2021-07-12

**Rating:** 7
**Confidence:** 4

**Summary:**

This paper extends recent information-theoretic approaches for computing model generalization to the meta-learning setting. They prove algorithm/data-dependent bounds for both the joint-training and support-query meta-learning strategies and instantiate the results for variants of gradient-based meta-learning with noise injection. Finally, they compute generalization bounds on synthetic and Omniglot data.

**Limitations And Societal Impact:**

Adequate.

**Main Review:**

Following the author response I have increased my score under the assumption that the additional discussions of closely related work and the new empirical generalization error results appear in the revised version.

# Original review

The extension of single-task information-theoretic generalization bounds to the meta-learning setting, together with the use of gradient incoherence-based bounds, seems like a meaningful and useful theoretical contribution. I lean towards acceptance of the paper, although I do believe there is not a significant amount of practical insight provided and the some experimental results and comparisons are missing, as discussed below; I think including them would significantly improve the paper.

### Strengths:
1. The paper proves new information-theoretic meta-learning generalization bounds. The bounds are algorithm/data-dependent and, when instantiated for MAML-like algorithms, rely on gradient incoherence rather than the gradient norm, allowing them to yield non-vacuous results.
2. Results are provided for both the joint training and support-query settings, which is an important direction for understanding meta-learning.
3. Experimental results demonstrate the usefulness of the bounds.
4. Code is released.

### Weaknesses:
1. I believe the paper would be significantly improved with more discussion comparing the results here directly to past work on meta-learning and information-theoretic generalization bounds. For the former, as discussed in the comments there are several recent papers that explicitly discuss the support-query approach and make direct comparisons to joint training. I believe there are useful comparisons to be made with the results in this paper. For the latter, while there is a comparison to single-task results, it is unclear how related the theory is. Are the proofs related or different from standard results and instantiations in this setting? I am not an expert there and so it is difficult to discern.
2. Experimentally, I believe it would be useful to compare the computed bounds with observed generalization error. This should certainly be doable for synthetic data and likely also for Omniglot. I believe this is important to give an understanding of how far away we are theoretically from practical performance.

### Comments:
17: This is not true of all few-shot learning algorithms, e.g. Reptile.
19: Papers by Denevi et al. (2019), Bai et al. (2021), and Saunshi et al. (2021) do discuss this in-depth.
102: The notation for a hypothesis w is lower-case on this line but upper-case in the inequality below. This inconsistency also appears elsewhere.
167: This reinforces the result of Bai et al. (2021), who showed that joint training is not asymptotically unbiased in the agnostic setting.
328: Doesn’t this make the meta-test distribution different from the meta-train distribution?

### References:
Bai, Chen, Zhou, Zhao, Lee, Kakade, Wang, Xiong. _How important is the train-validation split in meta-learning?_ ICML 2021.
Denevi, Ciliberto, Stamos, Pontil. _Learning to learn around a common mean._ NeurIPS 2019.
Saunshi, Gupta, Hu. _A representation learning perspective on the importance of train-validation splitting in meta-learning._ ICML 2021.

**Time Spent Reviewing:**

2

---

> ### Author Response · Authors · 2021-08-10
> **Response**
>
> We thank the reviewer for appreciating the contribution of our work, and we are grateful for the suggestions and additional related work which help improve our work.
>
> ### **Discussion with related work**
> We have carefully checked the mentioned related papers. We will add the discussions in our final version. Below are our discussions.
>
> - Recent papers that explicitly discuss the support-query approach.
>   * Denevi et al.[2018] first studied train-validation split for meta-learning in *biased linear regression* model. They proved a generalization bound and concluded that there exists a trade-off for train-validation split, which is consistent with Theorem 5.2 in our paper. Specifically, they constructed two datasets: For the simple unimodal distribution, the optimal split is $m_{tr}=0$. For the bimodal distribution, the optimal split is $m_{tr}\in(0, m-1]$.
>   * Bai et al. [2021] proposed a theoretical analysis of train-validation split in *linear centroid meta-learning* (parameter transfer). By comparing the train-val (alternate training) and train-train (joint training) method, they showed that train-validation split is necessary for the agnostic setting, where the train-val meta loss is an unbiased estimator w.r.t. the meta-test loss while the train-train loss is biased(consistent with our Theorem 5.1). When it is realizable (noiseless scenario), the train-train model can achieve better excess loss.
>   * Saunshi et al. [2021] analyze the train-valid splitting for *linear representation learning* (representation transfer). They proved that the train-validation split encourages learning a low-rank representation. In the noiseless setting, the train-val method already enables low-rank representation, so it's preferable to set a smaller train-split and larger validation-split.
>   * Our work focus on *general* settings with randomized algorithms and does not specify the form of base-learner and meta-learner, which can be applied in non-linear representation, non-linear classifier, and non-convex loss. Besides, the relations of our papers are as follows:
>     1. Our theory can recover the *stochastic* version of the above parameter and representation transfer settings. If we consider the linear model with $\mathcal{U}=\mathcal{W} \subseteq R^d$ and $P_{W|U}$ is approximated by a gaussian distribution $\mathcal{N}(U, I_{d})$, the problem is analogous to parameter-transfer meta-learning. If $\mathcal{U}\subseteq R^k, \mathcal{W} \subseteq R^{k+d}$ (where $U \in R^k$ is the shared representation parameter, $V \in R^d$ is the parameter of the linear classifier， $W=(U,V) \in R^{(k+d)}$ is the whole task parameter) and the prior of stochastic linear classifier $V $ is approximated by a gaussian distribution $\mathcal{N}(0,I_{d})$, the setting is similar to the representation transfer paradigm.
>     2. Since our bounds are based on the generic settings (flexible data distribution, algorithm, and loss choice), the two training modes are not directly comparable in our problem. However, we agree on the potential limit of joint training (asymptotically biased in the agnostic setting) and believe it is highly interesting to explore the specific conditions to understand the benefits and limitations of these training modes as the future work.
>
> - Discussion with other related information-theoretical results
>
>   Theorem 5.1 and 5.2 are based on the decoupling estimate lemma, which has been widely applied in information-theoretic single-task learning. However, our results are non-trivial extensions to the meta-learning scenarios. Specifically:
>   * In the PAC-Bayes framework, such as (Pentina et al.[2014] and Amit et al.[2018]), they separately upper-bounded the environment- and task-level error then combined the two-level risk via union bound. Following the two-level risk idea, Jose et al.[2020] extended to the information-theoretic meta-learning by introducing several unrealistic assumptions, leading to an uninterpretable and vacuous result in practice. On the contrary, we conducted the proofs differently. We consider the meta and task parameters as a joint term and exploit their probabilistic relation to obtain interpretable and practical meaningful bounds.
>   * Theorem 6.2 is crucial for obtaining non-vacuous MAML based theoretical bounds, which provides strong insights for understanding the generalization property of this well-established practice. However, the proof technique is a non-trivial extension of previous single-task learning settings, where they do not have a nested loop structure. We address this challenge by decomposing the global Markov structure to sub-structures through independent sampling and derive the upper bound for each substructure (Sec C.5 in appendix).
>
>
> ### **Comparison with the observed generalization error**
>
> We calculated the observed generalization error by evaluating the expected difference between the train loss and test loss.  And we list the results of synthetic data in the following tables:
>
>
>
> - $m_{tr}=8, m_{va}=8$:
>
> | epoch | 20 | 40 | 60 | 80 | 100 | 120 | 140 | 160 | 180 |
> | --- | --- | --- | --- | --- | --- | --- | --- | --- | --- |
> | Train-Test gap | 0.0697 | 0.0371 | 0.0908 | 0.0241 | 0.1072 | 0.1492 | 0.1775 | 0.1432 | 0.1581 |
> | G_norm | 6.009 | 7.506 | 8.807 | 9.856 | 10.643 | 11.558 | 12.386 | 13.275 | 14.014 |
> | G_inco (Ours) | 0.251 | 0.3462 | 0.4315 | 0.4976 | 0.5529 | 0.6112 | 0.6480 | 0.6983 | 0.7424 |
>
> - $m_{tr}=15, m_{va}=1$:
>
> | epoch | 20 | 40 | 60 | 80 | 100 | 120 | 140 | 160 | 180 |
> | --- | --- | --- | --- | --- | --- | --- | --- | --- | --- |
> | Train-Test gap |0.06857|0.03689|0.09728|0.0327|0.1113|0.1628|0.04986|0.06426|0.1674|
> | G_norm | 17.29|21.39|25.07|28.04|30.24|32.79|35.15|37.63|39.73|
> | G_inco (Ours) | 0.1774|0.2486|0.3127|0.3661|0.4074|0.4481|0.4817|0.5182|0.5468|
>
>
> - $m_{tr}=1, m_{va}=15$:
>
> | epoch | 20 | 40 | 60 | 80 | 100 | 120 | 140 | 160 | 180 |
> | --- | --- | --- | --- | --- | --- | --- | --- | --- | --- |
> | Train-Test gap |0.2389|0.06293|0.3155|0.1718|0.1822|0.2196|0.1687|0.1814|0.1923|
> | G_norm | 4.484|5.579|6.572|7.363|7.953|8.632|9.249|9.899|10.42|
> | G_inco (Ours)| 0.7801|1.048|1.286|1.45|1.617|1.76|1.89|2.018|2.149|
>
>
>
> Where Train-Test gap is the observed generalization error, G_inco is the whole gradient incoherence bound, i.e: $\sqrt{\frac{\sigma^2(\epsilon_U + \epsilon_W)}{nm_{va}}}$, G_norm is the corresponding bound w.r.t. gradient norm.
>
> Thus, we can see that the gradient-incoherence bound is much closer to the estimation of the actual gap but can be improved in the future.
>
> #### **References**
> Bai, Chen, Zhou, Zhao, Lee, Kakade, Wang, Xiong. How important is the train-validation split in meta-learning? ICML 2021.
>
> Denevi, Ciliberto, Stamos, Pontil. Learning to learn around a common mean. NeurIPS 2018.
>
> Saunshi, Gupta, Hu. A representation learning perspective on the importance of train-validation splitting in meta-learning. ICML 2021.
>
> Ron Amit and Ron Meir. Meta-learning by adjusting priors based on extended pac-bayes theory. InInternational Conference on Machine Learning, pages 205–214, 2018
>
> Anastasia Pentina and Christoph Lampert.  A pac-bayesian bound for lifelong learning.  In International Conference on Machine Learning, pages 991–999, 2014.
>
> Sharu Theresa Jose and Osvaldo Simeone. Information-theoretic generalization bounds for meta-learning and applications. arXiv preprint arXiv:2005.04372, 2020.

---

> > ### Comment · Reviewer_YsVe · 2021-08-17
> > **Follow-up**
> >
> > Thank you to the authors for responding to the reviews and for the additional experiments; I have updated my score. As a note on the statement *”Since our bounds are based on the generic settings (flexible data distribution, algorithm, and loss choice), the two training modes are not directly comparable in our problem.”*  I believe Bai et al. also compared convergence rates of non-equivalent notions of meta-test error, so there might still be a useful comparison to be made in this case.

---

> > > ### Author Response · Authors · 2021-08-17
> > > **Thanks for your feedback**
> > >
> > > We appreciate the reviewer's feedback and will update our final version according to your suggestions. We will further explore the two convergence rates then make a comparison.

---

### Official Review · Reviewer_kKnc · 2021-07-16

**Rating:** 7
**Confidence:** 2

**Summary:**

This paper provides information theoretic generalization bounds for meta-learning that 1) are algorithm and data dependent 2) don't depend on the norm of the gradients, but their sensitivity to the training data. This leads to stronger bounds than previous literature. Experiments on simulated data and Omniglot with the SGLD optimization algorithm confirm the theory.

**Limitations And Societal Impact:**

# Limitations #

The authors state that their work is limited to randomized algorithms.

# Societal impact #

I agree with the authors that there is no potential negative impact of the work, since it is primarily theoretical.

**Main Review:**

# Originality and significance #

This paper is primarily of theoretical interest to other researchers. It appears to be the second to provide information-theoretic data-dependent generalization bounds for meta-learning, after Jose and Simeone (2020). The bounds provided here appear to be tighter, as they are conditional on the meta-information $U$ for joint training and the support sets for alternate training. I cannot speak to the novelty of the mathematical tools used, as I do not work in this area.

# Quality #

Although I did not check the proofs in detail, the results appear to be sound. The experimental results validate the claim that their bounds are superior to those that rely on the norm of the gradient. However, I think in order to support other claims made in the paper, it would be helpful to 1) compare the bounds to the generalization error estimated using test data and 2) compare the bounds empirically to those of Jose and Simeone (2020).

# Clarity #

I found the paper to be well-written and clear. I would recommend that the authors move the legends of figures 2 and 3 below - the text is somewhat difficult to read.

# Update after reading response and other reviews #

I have decided to raise my score from 6 to 7, as the new empirical results addressed my concerns, with the understanding that they will be included in the final draft.

**Time Spent Reviewing:**

6

---

> ### Author Response · Authors · 2021-08-10
> **Response**
>
> We thank the reviewer for appreciating the contribution of our work, and we are grateful for the suggestions which help improve our work. Below are our responses to the main questions:
> ### **Discussion with Jose et al.[2020]**
>
> - Jose et al.[2020] adopted different and generally ***unrealistic*** assumptions to derive the theoretical results. Concretely:
>
>   * In joint-training (Eq (33) in Jose et al.[2020]), the task-level error w.r.t. base-learner $W$ is related to the unknown environment distribution $P_T$, which is hard to estimate from the observed data. In contrast, the task-level risk in our paper is associated with the distribution meta-parameter $U$, which can be evaluated efficiently. Besides, when $m\to\infty$ and the number of task $n$ is limited, their bound always has a non-zero term. This does not fit the reality since the new task already has enough samples to learn.
>   * In the alternate-training (meta train-validation) settings, they assumed the task parameters $W$ and $S^{va}$ are conditionally independent given $S^{tr}$ (Eq A(8) in their paper). This is an unrealistic condition in meta-learning since $W$ depends on the meta-parameter $U$, where $U$ is updated by $S^{va}_{1:n}$. As a result, if we set $m=1$ (each task has only one sample), then $n\to\infty$, the upper bound in Eq(3) Jose (2020) will converge to 0, which is problematic since task distribution can be arbitrary noisy and the task-level error (with one sample) can be quite large. Besides, this bound is irrelevant to the train validation split, which is inconsistent with the previous work such as Denevi et al.[2018] and Saunshi et al.[2021].
>
> - Therefore, our theoretical results are not directly comparable. Even if we ignore all these unrealistic theoretical assumptions and directly compare the results in Jose et al.[2020], their theoretical results in noisy iterative approaches still depend on the Lipschitz constant of the neural network (Eq~(45) in their paper), which is vacuous in deep learning.
>
> ### **Additional Empirical results**
> We have added the true meta generalization error(defined in our paper), which was estimated with test data (noted as the Train-Test gap in the table).
>
> In addition, we added the empirical estimation of Lipschitz bound in Eq (45)(Jose et al.[2020]), which is still much more significant than our gradient incoherence bound due to the large Lipschitz constant. Besides, their bound is independent of the train-validation split.
>
> The following tables compare the Train-Test gap with the gradient norm bound, the gradient incoherence bound and the bound of Jose et al.[2020] for different train-validation split settings in the synthetic data.
>
> - $m_{tr}=8, m_{va}=8$:
>
> | epoch | 20 | 40 | 60 | 80 | 100 | 120 | 140 | 160 | 180 |
> | --- | --- | --- | --- | --- | --- | --- | --- | --- | --- |
> | Train-Test gap | 0.0697 | 0.0371 | 0.0908 | 0.0241 | 0.1072 | 0.1492 | 0.1775 | 0.1432 | 0.1581 |
> | G_norm | 6.009 | 7.506 | 8.807 | 9.856 | 10.643 | 11.558 | 12.386 | 13.275 | 14.014 |
> | G_inco (Ours) | 0.251 | 0.3462 | 0.4315 | 0.4976 | 0.5529 | 0.6112 | 0.6480 | 0.6983 | 0.7424 |
> |Jose et al.[2020] |8.6074|12.1727|14.9085|17.2148|19.2468|21.0838|22.7731|24.3454|25.8222|
>
> - $m_{tr}=15, m_{va}=1$:
>
> | epoch | 20 | 40 | 60 | 80 | 100 | 120 | 140 | 160 | 180 |
> | --- | --- | --- | --- | --- | --- | --- | --- | --- | --- |
> | Train-Test gap |0.06857|0.03689|0.09728|0.0327|0.1113|0.1628|0.04986|0.06426|0.1674|
> | G_norm | 17.29|21.39|25.07|28.04|30.24|32.79|35.15|37.63|39.73|
> | G_inco (Ours) | 0.1774|0.2486|0.3127|0.3661|0.4074|0.4481|0.4817|0.5182|0.5468|
> | Jose et al.[2020] |8.6074|12.1727|14.9085|17.2148|19.2468|21.0838|22.7731|24.3454|25.8222|
>
>
> - $m_{tr}=1, m_{va}=15$:
>
> | epoch | 20 | 40 | 60 | 80 | 100 | 120 | 140 | 160 | 180 |
> | --- | --- | --- | --- | --- | --- | --- | --- | --- | --- |
> | Train-Test gap |0.2389|0.06293|0.3155|0.1718|0.1822|0.2196|0.1687|0.1814|0.1923|
> | G_norm | 4.484|5.579|6.572|7.363|7.953|8.632|9.249|9.899|10.42|
> | G_inco (Ours) | 0.7801|1.048|1.286|1.45|1.617|1.76|1.89|2.018|2.149|
> | Jose et al.[2020] |8.6074|12.1727|14.9085|17.2148|19.2468|21.0838|22.7731|24.3454|25.8222|
>
>
> #### **References**
> Sharu Theresa Jose and Osvaldo Simeone. Information-theoretic generalization bounds for meta-learning and applications. arXiv preprint arXiv:2005.04372, 2020.
>
> Denevi, Ciliberto, Stamos, Pontil. Learning to learn around a common mean. NeurIPS 2018.
>
> Saunshi, Gupta, Hu. A representation learning perspective on the importance of train-validation splitting in meta-learning. ICML 2021.

---

> > ### Comment · Reviewer_kKnc · 2021-08-17
> > **Thank you to the authors for your response**
> >
> > I have decided to raise my score from 6 to 7, as the updated empirical results addressed my questions.

---

> > > ### Author Response · Authors · 2021-08-17
> > > **Thank you for the feedback**
> > >
> > > We appreciate the reviewer's suggestions and will update our final version accordingly.

---

### Decision · Program_Chairs · 2021-09-27

**Decision:**

Accept (Spotlight)

**Comment:**

All reviewers agree that this paper is a solid work that contributes substantially to the theory of Meta-Learning. After the rebuttal, most reviewers found the response by the authors satisfactory and raised their score. As the AC of the paper, I also read the paper and found it novel and very well-written. I recommend acceptance of this work.